# Procurement Auctions with Predictions: Improved Frugality for Facility Location

**Eric Balkanski**
**Columbia University**
eb3224@columbia.edu

**Nicholas DeFilippis**
**New York University**
nad9961@nyu.edu

**Vasilis Gkatzelis**
**Drexel University,**
**Archimedes, Athena R.C., Greece**
gkatz@drexel.edu

**Xizhi Tan**[*]
**Stanford University**
xizhi@stanford.edu

## Abstract

We study the problem of designing procurement auctions for the strategic uncapacitated facility location problem: a company needs to procure a set of facility locations in order to serve its customers and each facility location is owned by a strategic agent. Each owner has a private cost for providing access to their facility (e.g., renting it or selling it to the company) and needs to be compensated accordingly. The goal is to design truthful auctions that decide which facilities the company should procure and how much to pay the corresponding owners, aiming to minimize the total cost, i.e., the monetary cost paid to the owners and the connection cost suffered by the customers (their distance to the nearest facility). We evaluate the performance of these auctions using the *frugality ratio*.

We first analyze the performance of the classic VCG auction in this context and prove that its frugality ratio is exactly 3. We then leverage the learning-augmented framework and design auctions that are augmented with predictions regarding the owners' private costs. Specifically, we propose a family of learning-augmented auctions that achieve significant payment reductions when the predictions are accurate, leading to much better frugality ratios. At the same time, we demonstrate that these auctions remain robust even if the predictions are arbitrarily inaccurate, and maintain reasonable frugality ratios even under adversarially chosen predictions. We finally provide a family of "error-tolerant" auctions that maintain improved frugality ratios even if the predictions are only approximately accurate, and we provide upper bounds on their frugality ratio as a function of the prediction error.

## 1 Introduction

When government agencies, retail chains, or banking institutions need to open new facilities or branches to better serve their customers, they face a challenging optimization problem: they want customers to be as close as possible to a facility, but opening new facilities can be very costly, and the cost can vary significantly by location. Deciding which locations to open therefore requires balancing the customers' *connection costs* (their distance to the nearest facility) against the facilities' *opening costs*. This optimization problem is known as the *uncapacitated facility location* (UFL) problem and it has received a lot of attention in prior work (e.g., [14, 21, 29, 36, 36, 40, 47]).

However, the vast majority of this prior work assumes that the designer has full information regarding both the connection and the opening costs, which is often unrealistic in practice. For example, each location may be owned by a different strategic agent who would need to be appropriately compensated

---

[*]This work was done while the author was a PhD student at Drexel University

39th Conference on Neural Information Processing Systems (NeurIPS 2025).

for the agency to open a facility there. Negotiating this compensation (i.e., the opening cost) is itself a strategic problem: the owner wishes to maximize it, while the agency wishes to minimize it. This additional obstacle is captured by the *strategic* UFL problem, where the opening cost of each location is private information held by the agent who own it [3, 19, 35, 48].

From the agency's perspective, rather than negotiating compensation separately with each owner, a more effective approach is to run a procurement auction. This encourages competition, leading to more efficient and cost-effective outcomes, which is one primary reason why the US government routinely uses procurement auctions [44]. To prevent manipulation, prior work on strategic UFL has focused on designing *truthful* auctions that incentivize agents to reveal their true opening costs [3, 19, 35, 48]. Our work continues in this direction.

Evaluating the performance of a procurement auction is complicated by the lack of a suitable benchmark. If there is little competition, the agency may have to pay higher compensation, whereas greater competition can reduce costs. The *frugality* literature [4, 48] addresses this by proposing the cost of the "second-best" solution as a benchmark (see Section 2). When this second-best cost is much higher than the optimum, the instance resembles a monopoly, and the agency must pay more. Conversely, if the second-best is close to the optimum, a well-designed auction can leverage competition for better outcomes. The *frugality ratio* quantifies how closely the auction's cost approaches this benchmark, and our main technical results are new frugality ratio bounds.

Despite its appeal, most prior work in frugal mechanism design is restricted to the adversarial framework, analyzing the auctions' performance under worst-case instances and assuming they have no information regarding the private costs. While this provides robust guarantees, it can be overly pessimistic, especially in practice, where agencies often have access to predictions regarding opening costs, derived from historical data, expert estimates, or data-driven models. This naturally raises the question: can we design mechanisms that exploit such predictions to improve frugality, while still remaining robust if the predictions turn out to be inaccurate?

Motivated by this, our work leverages the *learning-augmented framework*, which has recently spurred significant research on "algorithms with predictions" [37, 42], and more recently, on the design of truthful auctions and mechanisms augmented with predictions [1, 49]. In this framework, the designer is given an unreliable prediction (potentially from machine learning or historical data), and the goal is to achieve improved guarantees when the prediction is accurate (*consistency*), while maintaining strong guarantees even when the prediction is arbitrarily inaccurate (*robustness*).

## 1.1  Our Results

Our first result is a tight analysis of the frugality ratio of the classic truthful VCG auction (without predictions) for the strategic UFL problem. Prior to this result, the best known upper bound was $4$ by Talwar [48], and there were no known lower bounds. Using a tighter analysis and a matching lower bound, we prove that its frugality ratio is exactly $3$.

Then, rather than assuming that the auction has no information regarding the private opening cost $o_\ell$ at each location $\ell$, we consider *learning-augmented auctions* that are provided with a prediction $\hat{o}_\ell$ regarding this cost. Crucially, this prediction is unreliable and can be arbitrarily inaccurate.

Our second result is a new truthful procurement auction that takes a parameter $\epsilon \in (0, 2]$ as input and guarantees a frugality ratio of $1 + \epsilon$ whenever the prediction is accurate (the *consistency* guarantee), while simultaneously guaranteeing a frugality ratio of at most $\max\left(5, 3 + \frac{2}{\epsilon}\right)$, irrespective of how inaccurate the prediction may be (the *robustness* guarantee). Note that choosing a small constant $\epsilon$ allows us to guarantee a near-optimal frugality (arbitrarily close to $1$) when the prediction is accurate, while simultaneously guaranteeing a constant frugality even for adversarially chosen predictions. To achieve this guarantee, our learning-augmented auction uses the VCG mechanism on input that is scaled using the predictions. Specifically, if the opening cost reported by the owner at some location $\ell$ is exceeds the prediction $\hat{o}_\ell$, then it is scaled up even higher, reducing their chance of being selected. When the prediction is accurate, this limits the agents' ability to manipulate, leading to reduced costs.

For our third result, we design a learning-augmented procurement auction that is "error tolerant." We first define the prediction error as $\eta = \max_{\ell \in L} \max\left(\hat{o}_\ell/o_\ell, o_\ell/\hat{o}_\ell\right)$, i.e., the largest ratio between the predicted opening cost and the actual opening cost. Our auction is then provided with an error tolerance parameter, $\lambda > 1$, as input (as well as the parameter $\epsilon \in (0, 2]$), and as long as the error $\eta$ of

the prediction is at most $\lambda$, it guarantees a frugality ratio of $\eta(1+\lambda)+2\epsilon$. Even if the error exceeds the error tolerance threshold, i.e., $\eta > \lambda$, the frugality ratio is at most $\max\{2\lambda^4 + 3\lambda^2,\ 3 + \frac{2}{\epsilon}\}$.

## 1.2 Related Work

The strategic UFL problem was first studied by Archer and Tardos [3], who analyzed the structure of truthful mechanisms for this problem. The frugality of truthful mechanisms for the strategic UFL problem was studied by Talwar [48], who showed that the VCG auction has a frugality ratio of at most 4. Chen et al. [19] considered different social cost objectives for the special case where the facility locations coincide with the locations of the agents (i.e., each agent has a "dual-role" as a customer or facility operator). Chen et al. [19] and Li et al. [35] also considered the budget-feasible version of this problem, where the overall payment cannot exceed a budget. For related work on the non-strategic version of the UFL problem, see Appendix A.

The study of frugal mechanism design is motivated by the challenge of minimizing unnecessary overpayment while preserving truthfulness in procurement auctions and combinatorial team selection problems. This perspective has been applied across a variety of classic optimization settings. In network design, Archer and Tardos [4] and Talwar [48] initiated the investigation for path auctions, demonstrating that VCG mechanisms can incur large overpayments and establishing tight frugality bounds under the frugal solution benchmark. For coverage problems such as vertex cover and set cover, Elkind et al. [24] and subsequent works developed truthful mechanisms with frugality ratios that depend on structural parameters like graph degree or spectral properties, and also provided matching lower bounds in several cases. Frugality in $k$-flow and cut problems has also been explored, with mechanisms achieving constant-competitive frugality ratios relative to various benchmarks, including both disjoint-alternative and equilibrium-based definitions [17, 33, 34]. In some settings, such as matroid and spanning tree auctions, VCG mechanisms are provably frugal, achieving optimal or near-optimal frugality ratios [33, 48].

The design of learning-augmented mechanisms for settings involving strategic agents was initiated by Agrawal et al. [1] and Xu and Lu [49]. This line of work spans strategic facility location [1, 8, 12, 18, 31, 46, 49], strategic scheduling [7, 20, 49], auction design [9, 15, 16, 28, 38, 41, 49], bicriteria mechanism design for welfare and revenue trade-offs [6], graph problems with private input [22], distortion [13, 25], and equilibrium analysis [27, 32]. The work most closely related to this paper is [49], which studied frugal mechanism design with predictions in the context of path auctions. See Appendix A for a more extensive discussion on the line of work on algorithms with predictions.

## 2 Preliminaries

In the *Uncapacitated Facility Location (UFL) problem*, there is a set $U$ of users and a set $L$ of facilities. Each facility $\ell \in L$ has an opening cost $o_\ell$. Each user $u \in U$ and facility $\ell \in L$ have a connection cost $d(u, \ell)$. The connection costs are assumed to form a metric space, i.e., $d(x, y)$ is defined for any $x, y \in U \cup L$ and satisfies $d(x, x) = 0$, $d(x, y) \geq 0$ (non-negativity), $d(x, y) = d(y, x)$ (symmetry), and $d(x, z) \leq d(x, y) + d(y, z)$ (triangle inequality) for all $x, y, z \in U \cup L$. The cost of connecting a user $u \in U$ to a set of facilities $S \subseteq L$ is $\min_{\ell \in S} d(u, \ell)$, i.e., it is its distance to its closest facility in $S$. Given facilities $S \subseteq L$, we say that a user $u$ is assigned to facility $\ell$ if $\ell = \arg\min_{\ell \in S} d(u, \ell)$. The total connection cost incurred by a set of facilities $S \subseteq L$ is

$$d(U, S) = \sum_{u \in U} \min_{\ell \in S} d(u, \ell),$$

and its total cost $c(S)$ is the sum of its opening cost and total connection cost, i.e.,

$$c(S) = \sum_{\ell \in S} o_\ell + d(U, S).$$

The goal is to open a set of facilities $S \subseteq L$ that minimizes the total cost. Given $U, d$, and $\mathbf{o} = (o_\ell)_{\ell \in L}$, the optimal facility set is $\texttt{OPT}(U, \mathbf{o}, d) = \arg\min_{S \subseteq L} c(S)$.

**Strategic UFL.** In the strategic version of the UFL problem, each facility $\ell \in L$ is owned by a strategic agent and its opening cost $o_\ell$ is private (the connection costs are public information). An auction $\mathcal{M}(U, \mathbf{b}, d)$ for strategic UFL takes as input the set of users $U$ and the connection costs $d$,

and it asks the owner of each location $\ell \in L$ to report an opening cost $b_\ell$ (which we refer to as a bid), leading to a bid vector $\mathbf{b} = (b_\ell)_{\ell \in L}$. For simplicity, we write $\mathcal{M}(\mathbf{b})$ and $\texttt{OPT}(\mathbf{o})$ when $U$ and $d$ are clear from context. We also use $\mathbf{b}_{-\ell}$ to refer to the vector of all bids excluding $b_\ell$. We say that a facility $\ell$ misreports if its reported cost is not its true cost, i.e., $b_\ell \neq o_\ell$. The output $(S, \mathbf{p})$ of an auction consists of a set $S \subseteq L$ of facilities to open and a payment $p_\ell$ to each facility $\ell \in S$ for opening. The utility of a facility from output $(S, \mathbf{p})$ is

$$u_\ell(S, \mathbf{p}) = \begin{cases} p_\ell - o_\ell & \text{if } \ell \in S \\ 0 & \text{otherwise} \end{cases}.$$

An auction is *truthful* if it is a dominant strategy for every facility $\ell$ to report its true opening cost, i.e., $b_\ell = o_\ell$, irrespective of what the other facilities report. That is, for every $o_\ell, b_\ell, \mathbf{b}_{-\ell}, U, d$,

$$u_\ell(\mathcal{M}(o_\ell, \mathbf{b}_{-\ell})) \geq u_\ell(\mathcal{M}(b_\ell, \mathbf{b}_{-\ell})).$$

An auction $\mathcal{M}$ for UFL is *monotone* if for any facility $\ell \in L$ and any bid profile $\mathbf{b}_{-\ell}$ of the other facilities, the probability that $\ell \in S$, where $S$ is the facilities chosen by $\mathcal{M}(b_\ell, \mathbf{b}_{-\ell})$, is non-increasing in its bid $b_\ell$. In single-parameter environments such as strategic UFL, Myerson's Lemma characterizes truthful auctions:

**Lemma 2.1** (Myerson's Lemma). *An auction is truthful if and only if it is monotone. For any monotone auction, there exists a unique payment rule that ensures truthfulness, which can be computed explicitly.*

**Frugality and Efficiency.** The total cost incurred by an auction for outputting $(S, \mathbf{p})$ is the sum of the payments and the total connection cost, i.e.,

$$p(S, \mathbf{p}) = \sum_{\ell \in S} p_\ell + d(U, S).$$

The *frugal facility set* is the second-best solution, i.e.,

$$F(U, \mathbf{o}, d) = \underset{S \subseteq L \backslash \texttt{OPT}(U, \mathbf{o}, d)}{\arg \min} c(S).$$

The efficiency of an auction is evaluated by its *frugality ratio*, which is the worst-case ratio between its total cost and the cost of the frugal solution:

$$\texttt{frugality}(\mathcal{M}) = \max_{U, \mathbf{o}, d} \frac{p(\mathcal{M}(U, \mathbf{o}, d))}{c(F(U, \mathbf{o}, d))}.$$

**Learning-Augmented Framework.** In the *learning-augmented setting*, the auction is given predictions $\hat{\mathbf{o}}$ about the opening costs $\mathbf{o}$. We evaluate the performance of $\mathcal{M}$ using two metrics.

**Consistency:** The frugality ratio when the predictions are accurate ($\hat{\mathbf{o}} = \mathbf{o}$):

$$\texttt{consistency}(\mathcal{M}) = \max_{U, \mathbf{o}, d} \frac{p(\mathcal{M}(U, \mathbf{o}, d, \hat{\mathbf{o}} = \mathbf{o}))}{c(F(U, \mathbf{o}, d))}.$$

**Robustness:** The frugality ratio when the predictions can be arbitrarily wrong:

$$\texttt{robustness}(\mathcal{M}) = \max_{U, \mathbf{o}, d, \hat{\mathbf{o}}} \frac{p(\mathcal{M}(U, \mathbf{o}, d, \hat{\mathbf{o}}))}{c(F(U, \mathbf{o}))}.$$

## 3 Tight Frugality Bounds for the Vickrey-Clarke-Groves Auction

Our first main result establishes an improved frugality ratio of 3 for the VCG auction, which we start by describing in the context of the strategic UFL problem.

**The VCG auction.** Given an instance $(U, \mathbf{o}, d)$ of UFL, the set of facilities opened by the VCG auction is the optimal facility set $\texttt{OPT}(\mathbf{o})$. The VCG payment, also called threshold payment, to each opened facility $\ell \in \texttt{OPT}(\mathbf{o})$ is

$$p_\ell = c\big(\texttt{OPT}(\infty, o_{-\ell})\big) - c\big(\texttt{OPT}(0, \mathbf{o}_{-\ell})\big),$$

i.e., it is the difference between the total cost of the optimal solution when facility $\ell$ is infeasible ($o_\ell = \infty$) and when $\ell$ is free ($o_\ell = 0$). This payment can be alternatively defined as

$$p_\ell = \max\big\{ b \geq 0 : \ell \in \mathtt{OPT}(b, \mathbf{o}_{-\ell}) \big\},$$

i.e., the largest bid that facility $\ell$ can declare and remain in the optimal set $\mathtt{OPT}$ selected by VCG.

**Theorem 3.1.** *The frugality ratio of the VCG auction for the metric uncapacitated facility location problem is exactly* 3.

We first argue an upper bound of 3. At a high level, the proof bounds the total cost incurred by the VCG auction by (1) bounding the payments by twice the total cost of the frugal solution and (2) observing that the total connection cost of the optimal solution is bounded by its own total cost, which is in turn bounded by the total cost of the frugal solution. Combining these two facts immediately yields a 3-approximation. The payment bound is achieved by a "rerouting" argument: for each facility in the optimal set, we show how to reassign its users to other facilities, either within the optimal set itself or to those in the frugal solution, thereby upper-bounding its threshold payment by the corresponding rerouting cost.

*Proof of Theorem 3.1.* Let $\mathtt{OPT}$ and $F$ be the optimal and frugal solutions. We denote by $U_\ell \subseteq U$ the users assigned to facility $\ell$ according to $\mathtt{OPT}$. We first note that for any set of facilities $O_\ell \subseteq L \setminus \{\ell\}$ that does not contain facility $\ell$, we must have

$$p_\ell + \sum_{u \in U_\ell} d(u, \ell) \leq \sum_{f \in O_\ell \setminus \mathtt{OPT}} o_f + \sum_{u \in U_\ell} \min_{f' \in O_\ell} d(u, f'), \tag{1}$$

otherwise $O_\ell$ would have a smaller total cost than $\mathtt{OPT}$ when facility $\ell$ reports opening cost $p_\ell$, which would imply $p_\ell > \max\big\{ b \geq 0 : \ell \in \mathtt{OPT}(b, \mathbf{o}_{-\ell}) \big\}$ and would contradict the alternative definition of VCG.

The central part of the proof consists of defining some alternative solution $O_\ell$ and to reassign and reroute the users in $U_\ell$ to facilities in $O_\ell$ to bound $\sum_{u \in U_\ell} \min_{f' \in O_\ell} d(u, f')$. To define $O_\ell$ and this reassignment, we first need to introduce some notation. For each pair $(\ell \in \mathtt{OPT}, f \in F)$, we let $U_{\ell,f} \subseteq U$ be the set of users assigned to $\ell$ in the optimal solution $\mathtt{OPT}$ and to $f$ in the frugal solution $F$. For each $f \in F$, let $\mathtt{OPT}_f = \{\ell \in \mathtt{OPT} : |U_{\ell,f}| > 0\}$ and $x_f(\ell)$ be the ranking of $\ell \in \mathtt{OPT}_f$ among all other facilities in $\mathtt{OPT}_f$ in terms of the size of $|U_{\ell,f}|$, where ties are broken arbitrarily but consistently. This definition implies that $x_f^{-1}(j) \in \mathtt{OPT}_f$ is the facility $\ell$ with the $j^{th}$ smallest non-negative $|U_{\ell,f}|$ and we thus have $0 < |U_{x_f^{-1}(1),f}| \leq \cdots \leq |U_{x_f^{-1}(|\mathtt{OPT}_f|),f}|$.

We define the alternate solution to be $O_\ell = \mathtt{OPT} \setminus \{\ell\} \cup \{f \in F : x_f(\ell) = |\mathtt{OPT}_f|\}$. In this solution, facility $\ell$ is removed from $\mathtt{OPT}$ and the facilities $f \in F$ such that $|U_{\ell,f}|$ is largest among $\{|U_{\ell',f}|\}_{\ell' \in \mathtt{OPT}}$ are added. We then define a mapping $\pi_f : \mathtt{OPT}_f \to \mathtt{OPT}_f \cup \{f\}$ that reassigns users in $U_{\ell,f}$ that are assigned to facility $\ell$ according to $\mathtt{OPT}$ to a new facility in $\mathtt{OPT}_f \cup \{f\}$. This reassignment is defined as

$$\pi_f(\ell) = \begin{cases} x_f^{-1}(x_f(\ell) + 1) & \text{if } x_f(\ell) < |\mathtt{OPT}_f| \\ f & \text{if } x_f(\ell) = |\mathtt{OPT}_f| \end{cases}.$$

In words, $\pi_f(\ell)$ maps facility $\ell$ to the facility ranked immediately after $\ell$ with respect to $f$; if $\ell$ is ranked last, then $\pi_f(\ell) = f$. Since $\cup_{f:\ell \in \mathtt{OPT}_f} U_{\ell,f} = U_\ell$, we have that

$$\sum_{u \in U_\ell} \min_{f' \in O_\ell} d(u, f') \leq \sum_{f:\ell \in \mathtt{OPT}_f} \min_{f' \in O_\ell} d(U_{\ell,f}, f') \leq \sum_{f:\ell \in \mathtt{OPT}_f} d(U_{\ell,f}, \pi_f(\ell)). \tag{2}$$

Next, if $x_f(\ell) < |\mathtt{OPT}_f|$, then $|U_{\ell,f}| \leq |U_{\pi_f(\ell),f}|$ by the definition of $\pi_f$. Thus, combined with the triangle inequality, we get that if $x_f(\ell) < |\mathtt{OPT}_f|$, then

$$d(U_{\ell,f}, \pi_f(\ell)) \leq \sum_{f:\ell \in \mathtt{OPT}_f} d(U_{\ell,f}, f) + d\big(f, U_{\pi_f(\ell),f}\big) + d\big(U_{\pi_f(\ell),f}, \pi_f(\ell)\big). \tag{3}$$

Informally, this last inequality corresponds to rerouting the users in $U_{\ell,f}$ to $\pi_f(\ell)$ by going through $f$ and then $U_{\pi_f(\ell),f}$. This rerouting is the key component of this proof, see Figure 1 for an illustration.

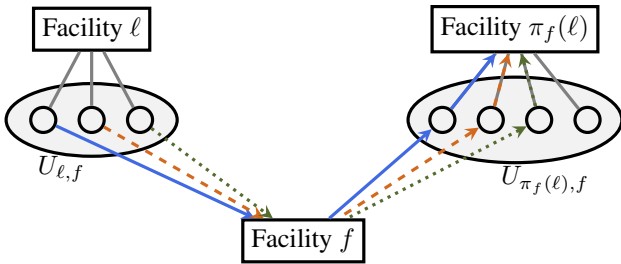

Figure 1: Illustration of how users in $U_{\ell,f}$ are rerouted to facility $\pi_f(\ell)$. Each color (and line pattern) denotes the path taken by a distinct user $u \in U_{\ell,f}$. The total connection-cost along each set of colored edges gives an upper bound on the payment to facility $\ell$ for serving that subset of users.

We obtain that

$$
p(\texttt{OPT}) = \sum_{\ell \in \texttt{OPT}} \left( p_\ell + \sum_{u \in U_\ell} d(u,\ell) \right) \qquad\qquad \text{by def. of } p
$$

$$
\leq \sum_{\ell \in \texttt{OPT}} \left( \sum_{f: x_f(\ell) = |\texttt{OPT}_f|} o_f + \sum_{u \in U_\ell} \min_{f' \in O_\ell} d(u,f') \right) \qquad\qquad \text{by (1)}
$$

$$
\leq \sum_{\ell \in \texttt{OPT}} \left( \sum_{f: x_f(\ell) = |\texttt{OPT}_f|} o_f + \sum_{f: \ell \in \texttt{OPT}_f} d(U_{\ell,f}, \pi_f(\ell)) \right) \qquad\qquad \text{by (2)}
$$

$$
= \sum_{f \in F} \left( o_f + \sum_{\ell \in \texttt{OPT}_f} d(U_{\ell,f}, \pi_f(\ell)) \right)
$$

$$
= \sum_{f \in F} \left( o_f + d\left( U_{x_f^{-1}(|\texttt{OPT}_f|),f}, f \right) + \sum_{j=1}^{|\texttt{OPT}_f|-1} d\left( U_{x_f^{-1}(j),f}, x_f^{-1}(j+1) \right) \right)
$$

$$
\leq \sum_{f \in F} \left( o_f + d\left( U_{x_f^{-1}(|\texttt{OPT}_f|),f}, f \right) \right.
$$
$$
\left. + \sum_{j=1}^{|\texttt{OPT}_f|-1} \left( d(U_{x_f^{-1}(j),f}, f) + d(f, U_{x_f^{-1}(j+1),f}) \right.\right.
$$
$$
\left.\left. + d(U_{x_f^{-1}(j+1),f}, x_f^{-1}(j+1)) \right) \right) \qquad\qquad \text{by (3)}
$$

$$
\leq \sum_{f \in F} \left( o_f + \sum_{\ell \in \texttt{OPT}_f} (d(U_{\ell,f}, f) + d(f, U_{\ell,f}) + d(U_{\ell,f}, \ell)) \right)
$$

$$
= \left( \sum_{f \in F} o_f + 2 \sum_{f \in F} \sum_{\ell \in \texttt{OPT}_f} d(U_{\ell,f}, f) \right) + \sum_{\ell \in \texttt{OPT}} \sum_{f: \ell \in \texttt{OPT}_f} +d(U_{\ell,f}, \ell)
$$

$$
\leq 2 \cdot c(F) + c(\texttt{OPT})
$$

$$
\leq 3 \cdot c(F).
$$

We show that this upper bound of 3 is tight for VCG by providing a simple tree-metric instance. Due to space limitations, we defer its analysis to Appendix B. □

# 4 Learning-Augmented Frugal Auction

We now consider the strategic UFL problem within the learning-augmented framework, where the auction has access to potentially inaccurate predictions about the true opening costs of the facilities. Specifically, for each facility $\ell \in L$, a prediction $\hat{o}_\ell$ is provided for its true opening cost $o_\ell$. Our objective is to leverage this unreliable information to design truthful auctions that minimize the frugality ratio while remaining robust against inaccurate predictions.

To achieve this, we propose PREDICTEDLIMITS, described in Auction 1. This auction first identifies the optimal solution based on the predicted costs, denoted by $\hat{\texttt{OPT}}$. It then modifies the cost function by scaling the actual opening costs of facilities in $\hat{\texttt{OPT}}$ when their true costs exceed the predicted opening costs. Finally, the auction selects the solution that minimizes this scaled cost function. We note that,

---

**Auction 1:** PREDICTEDLIMITS

**Input:** Set of clients $U$, set of facilities $L$, parameter $\epsilon$, predicted opening costs $\hat{o}_\ell$ for all $\ell \in L$
**Output:** Selected facility subset $S^*$ and threshold payments for each facility $\ell \in S^*$
Compute $\hat{\texttt{OPT}} \leftarrow \arg\min_{S \subseteq L} \left( d(U, S) + \sum_{l \in S} \hat{o}_\ell \right)$;
Define the modified cost function for any subset $S \subseteq L$:

$$o'_\ell(S) \leftarrow \begin{cases} \frac{2}{\epsilon} \cdot o_\ell, & \text{if } S = \hat{\texttt{OPT}} \text{ and } o_\ell > \hat{o}_\ell \\ o_\ell, & \text{otherwise} \end{cases}$$

Compute:

$$S^* \leftarrow \arg\min_{S \subseteq L} \left( d(U, S) + \sum_{\ell \in S} o'_\ell(S) \right)$$

**return** $S^*$ and threshold payments for each $\ell \in S^*$;

---

to achieve the claimed results, our auction only needs access to the membership of the predicted optimal solution $\hat{\texttt{OPT}}$ and to the opening costs of those facilities. We show that PREDICTEDLIMITS can achieve near-optimal consistency (arbitrarily close to 1) while still maintaining competitive robustness.

**Theorem 4.1.** *Given any $\epsilon \in [0, 2]$ as input,* PREDICTEDLIMITS *is truthful, $(1 + \epsilon)$-consistent, and $\max\left(5, 3 + \frac{2}{\epsilon}\right)$-robust.*

At first glance, this auction may seem counterintuitive and differs from existing mechanisms that also use predictions to scale costs (e.g, [7, 49]). Instead of scaling down the opening costs of facilities in the predicted optimal solution, a set that would normally be favored for minimizing the overall social cost, the auction scales these costs upwards, making the predicted optimal solution less appealing. The primary rationale for this upward scaling is to restrict the potential overbidding the bidder can claim while remaining in the optimal solution, which leads to a high threshold price. By amplifying any over-reported opening costs over the predicted ones, the auction gains tighter control over facility payments, which can lead to a lower frugality ratio when the predictions are accurate.

Importantly, the scaling applied by our auction is *solution-dependent* rather than facility-dependent. Specifically, the opening cost scaling is activated only when evaluating the predicted optimal solution $\hat{\texttt{OPT}}$ as a complete set. For any other subset, the opening costs of individual facilities remain unscaled. This distinction is crucial for our robustness analysis, as it ensures that the adverse effects of inaccurate predictions are confined to the evaluation of the single predicted optimal solution, leaving alternative solutions unaffected. We defer the truthfulness proof to Appendix C.1. We first provide the consistency analysis of Auction PREDICTEDLIMITS. We then provide the robustness guarantee of the auction, which holds regardless of the quality of the predictions.

**Lemma 4.2.** *Given any $\epsilon \in [0, 2]$ as input,* PREDICTEDLIMITS *achieves a consistency of $1 + \epsilon$.*

*Proof.* Recall that consistency analysis assumes the predictions are correct, i.e., $\hat{o}_\ell = o_\ell$ for all facilities $\ell \in L$. By the definition of Auction 1, if predictions are accurate, the condition $o_\ell > \hat{o}_\ell$ for $\ell \in \texttt{OPT}$ (which equals $\hat{\texttt{OPT}}$ here) is never met. Therefore, no costs are scaled, and the auction selects the true optimal solution, i.e., $S^* = \texttt{OPT}$. We adopt the same definitions of $\texttt{OPT}_f$ and $\pi_f(\ell)$ as in the analysis of VCG (proof of Theorem 3.1).

We now provide an upper bound for the payment $p_\ell$ to the agent controlling facility $\ell \in \texttt{OPT}$. Similarly to VCG, we will upper-bound the payment by considering the cost of rerouting. The difference is that when the reported bid exceeds $\hat{o}_\ell$, the scaling would apply. Formally, for any set of facilities $O_\ell \subseteq L \setminus \{\ell\}$ not containing $\ell$:

$$\frac{2}{\epsilon} p_\ell + \sum_{u \in U_\ell} d(u, \ell) \leq \sum_{f' \in O_\ell \setminus \texttt{OPT}} o_{f'} + \sum_{u \in U_\ell} \min_{f' \in O_\ell} d(u, f'). \tag{4}$$

Note that the scaling affects only the opening costs of the facilities in $\hat{\texttt{OPT}}$. Hence the right-hand side of (4) remains unchanged for all $(\ell, f)$ pairs relative to the VCG analysis, as it represents cost of alternative solutions not involving $\ell$'s potentially scaled cost. Therefore, the bounds from Equations (2) and (3) continue to hold. Following the same steps as in the VCG proof (Theorem 3.1), we obtain:

$$\sum_{\ell \in \texttt{OPT}} \left( \frac{2}{\epsilon} p_\ell + \sum_{u \in U_\ell} d(u, \ell) \right) \leq \left( \sum_{f \in F} o_f + 2 \sum_{f \in F} \sum_{\ell \in \texttt{OPT}_f} d(U_{\ell, f}, f) \right) + \sum_{\ell \in \texttt{OPT}} \sum_{f : \ell \in \texttt{OPT}_f} d(U_{\ell, f}, \ell)$$

$$\Rightarrow \quad \frac{2}{\epsilon} \sum_{\ell \in \texttt{OPT}} p_\ell \leq 2 \sum_{f \in F} o_f + 2 \sum_{f \in F} \sum_{\ell \in \texttt{OPT}_f} d(U_{\ell, f}, f) \quad \Rightarrow \quad \sum_{\ell \in \texttt{OPT}} p_\ell \leq \epsilon \cdot c(F).$$

Together with the fact that the connection cost of the optimal solution is at most $c(F)$, we conclude that the total auction cost $\sum_{\ell \in \texttt{OPT}} p_\ell + d(U, \texttt{OPT})$ is at most $(1 + \epsilon) c(F)$. $\qquad \square$

We now move on to the robustness analysis. We begin with the simpler case in which the auction still outputs the optimal solution of the instance, i.e., $S^* = \texttt{OPT}$. All missing proofs of the section can be found in Appendix C.2.

**Lemma 4.3.** *Given any instance $(U, \mathbf{o}, d)$ and any $\epsilon \leq 2$, let $\texttt{OPT}$ and $F$ be the optimal solution and frugal solution, respectively. If auction 1 outputs $S^* = \texttt{OPT}$, then we have:*

$$\sum_{\ell \in S^*} p_\ell + d(U, S^*) \leq \left( 1 + \frac{2}{\epsilon} \right) \sum_{\ell \in F} o_\ell + 3 \, d(U, F)$$

The proof is deferred to Appendix C.2.1. We now address the case where scaling alters the output set, i.e., $S^* \neq \texttt{OPT}$. This case is particularly challenging because $S^*$ may contain facilities belonging to both the frugal solution and the optimal solution, making it difficult to bound the payment. To tackle this, we need a different accounting argument than the ones used for the VCG and consistency analyses. Specifically, we will analyze payments for facilities in the output solution $S^*$ differently, based on whether they are part of the frugal solution $F$. We begin by establishing a bound for payments to non-frugal facilities in $S^*$ (i.e., those in $S^* \setminus F$) with the following lemma.

**Lemma 4.4.** *Given any instance $(U, \mathbf{o}, d)$, and any $\epsilon \leq 2$, let $\texttt{OPT}$ and $F$ be the optimal solution and frugal solution, respectively. Let $S^* \neq \texttt{OPT}$ be the output of Auction 1, and let $S = S^* \setminus F$. Then:*

$$\sum_{\ell \in S} p_\ell \leq \sum_{\ell \in F} o_\ell + 2d(U, F).$$

The proof is deferred to Appendix C.2.2. Then, we bound the payment to the output solution that is also part of the frugal solution.

**Lemma 4.5.** *Given any instance $(U, \mathbf{o}, d)$, and any $\epsilon \leq 2$, let $\texttt{OPT}$ and $F$ be the optimal solution and frugal solution, respectively. Let $S^* \neq \texttt{OPT}$ be the output of Auction 1, and $S^f = S^* \cap F$. Then:*

$$\sum_{\ell \in S^f} p_\ell \leq \max \left( 2, \frac{2}{\epsilon} \right) \cdot \left[ \sum_{\ell \in F} o_\ell + d(U, F) \right].$$

*Proof.* The approach considers rerouting users currently assigned to a facility $f \in S^f$ back to their corresponding facility $\ell$ in the true optimal solution $\texttt{OPT}$. This facility $j \in \texttt{OPT}$ might not be included in the output set $S^*$, and if $\ell \in \texttt{OPT} = \hat{\texttt{OPT}}$, its opening cost $o_\ell$ might be effectively scaled

upwards to $o'_j$ in the auction's payment calculation if it was under-predicted ($o_\ell > \hat{o}_\ell$). The analysis mirrors the VCG structure but with the roles of the chosen solution ($S^f$) and the alternative (OPT) flipped. Define $U_\ell$, $U_{f,\ell}$, and $\pi_\ell(f)$ as in the proof of Theorem 3.1, replacing OPT with $S^f$. Define $S^f_\ell = \{\ell \in S^f : |U_{f,\ell}| > 0\}$.

Case 1: The optimal facility $\ell \notin S^*$. By the same rerouting argument as VCG, inequalites (2) and (3) hold with OPT replaced by $S^f$ and the opening cost replaced by the scaled opening cost.

Case 2: The optimal facility $\ell \in S^*$. Here, users $U_{f,\ell}$ can be rerouted directly to $\ell$ within the output set $S^*$, so the cost associated with rerouting users $U_{f,\ell}$, denoted obeys

$$p_\ell + \sum_{u \in U_\ell} d(u, \ell) \leq d(U_{f,\ell}, \ell).$$

This bound is better than inequalities (2) and (3) from Case 1.

Summing over all agents $f \in S^f$ and assuming the worst-case where every cost gets scaled we get:

$$\sum_{\ell \in S^f} \left( p_\ell + \sum_{u \in U_\ell} d(u, \ell) \right) = \sum_{\ell \in \text{OPT}} \sum_{f \in S^f_\ell} \left( p_\ell + \sum_{u \in U_\ell} d(u, \ell) \right)$$

$$\leq \sum_{\ell \in \text{OPT}} \left( \frac{2}{\epsilon} o_\ell + 2 \sum_{f \in S^f_\ell} d(U_{\ell,f}, f) \right) + \sum_{\ell \in \text{OPT}} \sum_{f : \ell \in \text{OPT}_f} d(U_{\ell,f}, \ell)$$

$$\Rightarrow \quad \sum_{\ell \in S^f} p_\ell \leq \sum_{\ell \in \text{OPT}} \frac{2}{\epsilon} o_\ell + 2 \sum_{\ell \in \text{OPT}} \sum_{f \in S^f_\ell} d(U_{\ell,f}, f)$$

$$\Rightarrow \quad \sum_{\ell \in \text{OPT}} p_\ell \leq \max\left(\frac{2}{\epsilon}, 2\right) c(F) = \max\left(\frac{2}{\epsilon}, 2\right) \left[\sum_{\ell \in F} o_\ell + d(U, F)\right]. \qquad \square$$

By combining Lemma 4.3, Lemma 4.4, and Lemma 4.5, we get the robustness of Auction PREDICT-EDLIMITS. The proof is deferred to Appendix C.2.3.

**Lemma 4.6.** *Given any* $\epsilon \in [0, 2]$ *as input,* PREDICTEDLIMITS *achieves a robustness of* $\max\left(5, 3 + \frac{2}{\epsilon}\right)$.

## 5 The Error-Tolerant Scaled VCG

Auction PREDICTEDLIMITS achieves $(1 + \epsilon)$-consistency in its frugality ratio when provided with perfect predictions of opening costs, and maintains a $\max\left(5, 3 + \frac{2}{\epsilon}\right)$-robustness guarantee in adversarial scenarios. These existing bounds, however, capture only the extremes of prediction quality. In this section, we extend the auction from the previous section to achieve an improved frugality ratio, not only for accurate predictions but also for approximately accurate predictions, thereby offering a more granular performance characterization.

We denote the prediction error as $\eta \geq 1$, where $\eta$ is defined as the largest ratio between the predicted opening cost and the actual opening cost, i.e.,

$$\eta = \max_{\ell \in L} \max\left(\frac{\hat{o}_\ell}{o_\ell}, \frac{o_\ell}{\hat{o}_\ell}\right).$$

The auction, ERRORTOLERANT, formally defined in Auction 2, extends PREDICTEDLIMITS by incorporating an error-tolerance parameter $\lambda \geq 0$. Recall that the original auction selects the predicted-optimal facility set $\hat{\text{OPT}}$ and, whenever a facility's true opening cost exceeds its prediction, inflates that cost by a factor of $2/\epsilon$; otherwise, it leaves opening costs unchanged.

ERRORTOLERANT is designed to preserve this behavior not only under exact predictions but also when predictions are accurate up to a factor of $\lambda$, with one key enhancement. To ensure that $\hat{\text{OPT}}$ remains the chosen set under low prediction error, we replace the overscaling threshold $\hat{o}_\ell$ with $\lambda \hat{o}_\ell$. In addition, if every facility in $\hat{\text{OPT}}$ reports a true cost below this new threshold, we apply a uniform

**Auction 2:** ERRORTOLERANT

**Input:** Clients $U$, facilities $L$, predicted costs $\hat{o}_\ell$ for all $\ell \in L$, parameter $\epsilon \le 2$, error-tolerance $\lambda \ge 0$

**Output:** Chosen facilities $S^*$ and threshold payments for each $\ell \in S^*$

$$\hat{\text{OPT}} \leftarrow \arg\min_{S \subseteq L}\Big(d(U,S) \ + \ \sum_{\ell \in S}\hat{o}_\ell\Big);$$

**if** $\forall \ell \in \hat{\text{OPT}}$, $o_\ell \le \lambda \hat{o}_\ell$ **then**

$$\text{Define} \quad c_\lambda(S) \ = \ \begin{cases} \dfrac{1}{\lambda^2}\Big(d(U,S) \ + \ \sum_{\ell \in S}o_\ell\Big), & S = \hat{\text{OPT}}, \\[2ex] d(U,S) \ + \ \sum_{\ell \in S}o_\ell, & \text{otherwise.} \end{cases}$$

**else**

$$\text{Define for each } \ell \text{ and } S \quad o'_\ell(S) \ = \ \begin{cases} \frac{2}{\epsilon}\,o_\ell, & S = \hat{\text{OPT}} \text{ and } o_\ell > \lambda\hat{o}_\ell, \\[1ex] o_\ell, & \text{otherwise,} \end{cases}$$

$$\text{and let} \quad c_\lambda(S) \ = \ d(U,S) \ + \ \sum_{\ell \in S}o'_\ell(S).$$

$$S^* \ \leftarrow \ \arg\min_{S \subseteq L}c_\lambda(S),$$

**return** $S^*$ *and threshold payments for each* $\ell \in S^*$;

---

down-scaling to the entire solution cost (both opening and connection costs). This extra adjustment guarantees that, when the prediction error is at most $\lambda$, the auction outputs $\hat{\text{OPT}}$. Aside from these two modifications, ERRORTOLERANT follows PREDICTEDLIMITS exactly. It achieves an approximation guarantee of $\eta\,(1+\lambda)+2\epsilon$ when $\eta \le \lambda$, and maintains a frugality bound of $\max(2\lambda^4+3\lambda^2, 3+2/\epsilon)$ for arbitrary error $\eta$.

**Theorem 5.1.** *Auction* ERRORTOLERANT *is truthful, and given parameters* $\epsilon \in (0,2]$ *and* $\lambda > 1$*, it achieves the following frugality ratio, where* $\eta$ *is the error of the prediction:*

$$\begin{cases} \eta(1+\lambda)+2\epsilon, & \text{if } \eta \le \lambda, \\[2ex] \max\Big\{2\lambda^4+3\lambda^2,\ 3+\frac{2}{\epsilon}\Big\}, & \text{if } \eta > \lambda. \end{cases}$$

Due to space limitations, we defer the analysis to Appendix E.

## 6  Conclusion and Open Problems

In this work, we studied the design of frugal procurement auctions for the strategic uncapacitated facility location problem. We first established a tight frugality ratio of 3 for the classic VCG auction. Prior to our work, the best known upper bound for VCG was 4 [48] and there was no known lower bound. We then considered the problem in the learning-augmented framework where the auction is provided with predictions regarding the costs of the facilities. We designed a novel truthful auction that achieves a frugality ratio of $1 + \epsilon$ when the predictions are accurate, while maintaining a constant-factor robustness guarantee even when the predictions are arbitrarily wrong.

Although the focus of our work is on the information-theoretic limitations of truthful auctions, we note that our proposed mechanisms can be formulated as integer programs (IPs). Please see Appendix F for the IP implementation.

Our work leaves several exciting directions open. In the classic setting, while we show that VCG's frugality ratio is exactly 3, the optimal frugality ratio achievable by a truthful auction remains unknown. An important question is whether VCG is optimal, or if another design can achieve a better frugality ratio. Thus, establishing a tight lower bound for this problem is a key next step. In the learning-augmented setting, an open problem is to determine the optimal trade-off between consistency and robustness, thereby characterizing the Pareto-optimal frontier. Furthermore, exploring more nuanced definitions of prediction error beyond magnitude—such as the number of mispredictions or the recent "mostly and approximately correct" (MAC) framework [12]—is a promising direction. Analyzing mechanisms under such models could lead to auctions that degrade more gracefully given imperfect, yet informative, predictions.

## Acknowledgments and Disclosure of Funding

Eric Balkanski was supported by NSF grants CCF-2210501 and IIS-2147361. Nicholas DeFilippis was supported by an NSF Graduate Research Fellowship under Grant No. DGE-2039655. Vasilis Gkatzelis and Xizhi Tan were supported by NSF CAREER award CCF-2047907 and NSF grant CCF-2210502. Any opinion, findings, and conclusions or recommendations expressed in this material are those of the authors and do not necessarily reflect the views of the National Science Foundation.

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

## A Extended Related Work

**Algorithms with Predictions.** In recent years, the learning-augmented framework has become a leading paradigm for algorithm design and analysis. For an overview of its early developments, see [43], and for a comprehensive, up-to-date survey, consult [37]. By incorporating (potentially imperfect) predictions, this framework overcomes the limitations of classical worst-case analyses. Indeed, over the past five years, hundreds of papers have revisited core algorithmic problems through this lens, with prominent examples including online paging [39], scheduling [45], covering and knapsack-constrained optimization [10, 30], Nash social welfare maximization [11], variations of the secretary problem [2, 23, 26], and a variety of graph-based challenges [5].

**UFL.** The metric Uncapacitated Facility Location (UFL) problem is a fundamental NP-hard optimization problem, widely studied due to its theoretical significance and practical relevance in operations research and algorithmic game theory. Early foundational results established the APX-hardness of the problem, showing that it cannot be approximated within a factor better than $1.463$ unless $P = NP$ [29]. The first constant-factor approximation was provided by Shmoys et al. [47], who achieved a ratio of approximately $3.16$. Subsequent improvements significantly narrowed the approximation gap, with notable progress by Chudak and Shmoys [21] to $1.736$ and Mahdian et al. [40] to $1.52$-approximation via sophisticated linear-programming techniques. Byrka and Aardal [14] later improved this bound to a $1.50$-approximation using a novel bifactor algorithm . The current state-of-the-art algorithm, due to Li [36], achieves an approximation ratio of $1.488$, leaving a small gap relative to the known hardness bound.

## B Tightness of the Frugality Bound for VCG

We now show that the frugality ratio of 3 for VCG is tight by exhibiting the following instance.

**Theorem B.1.** *The frugality ratio of VCG auction is at least* $3 - \frac{6}{|L|+1}$, *even for tree metrics. Hence, in the worst case (as* $|L| \to \infty$), *the frugality ratio approaches* $3$.

*Proof.* Consider the tree metric defined by facilities $L = \{\ell_0, \dots, \ell_k\}$, users $U = \{u_1, \dots, u_k\}$, and edges $E = \{\{\ell, u_i\}\}_{i=1}^k \cup \{\{u_i, \ell_0\}\}_{i=1}^k$ (see Figure B for an illustration). Facility $\ell_0$ has an opening cost $o_{\ell_0} = 2$ and the other facilities have an opening cost of $0$. The distance of from the central facility $\ell_0$ to any other facility $\ell$ is $2$ and $d(\ell, \ell_j) = 4$ for all pairs of distinct $i, j > 0$. Finally, we have one user $u_i$ in the midway of $\ell$ and $\ell_0$.

For the given instance, the optimal solution $\texttt{OPT}(U, \mathbf{o}, d)$ is to open all of the peripheral facilities $\{\ell_1, \dots, \ell_k\}$ for a total cost of $k$. Consequently, the frugal solution $F(U, o) = \{\ell_0\}$ is to open the central facility for a total cost of $k + 2$.

Now consider the payment made by VCG. Note that for all $i \in \{1, \dots, k\}$, if the opening cost of $\ell$ is $o_\ell = \infty$ (and the other opening costs do not change), then user $u_i$ must connect either to the central

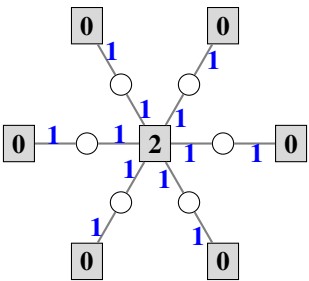

Figure 2: Illustration of the lower bound instance where each square represents a facility location with its opening cost, and the circles represent users.

facility $\ell_0$ (which has opening cost 2) or to another peripheral facility (which increases the connection cost of $u_i$ by 2), so the solution $\mathtt{OPT}(\infty, \mathbf{o}_{-l})$ is either $\{\ell_0\}$ or $\{\ell_1, \ldots, \ell_{i-1}, \ell_{i+1}, \ldots, \ell_k\}$ and $c(\mathtt{OPT}(\infty, \mathbf{o}_{-l})) = k + 2$. Thus, for each peripheral facility $\ell$ with $i \in \{1, \ldots, k\}$, the VCG payment is

$$p_{\ell_i} = c(\mathtt{OPT}(\infty, \mathbf{o}_{-l})) - c(\mathtt{OPT}(0, \mathbf{o}_{-l})) = (k+2) - k = 2.$$

Therefore, the total payments by the VCG auction to the facilities $\mathtt{OPT}(U, \mathbf{o}) = \{\ell_1, \ldots, \ell_k\}$ that it selects is $2k$ which, combined with a total user connection costs of $k$, leads to a total cost of $3k$. Therefore, the frugality ratio for this instance is

$$\frac{3k}{c(F(U,o))} = \frac{3k}{k+2} = \frac{3(|L|-1)}{|L|+1} = \frac{3(|L|+1)-6)}{|L|+1} = 3 - \frac{6}{|L|+1}. \quad \square$$

## C  Missing Proofs from Section 4

### C.1  Truthfulness of PREDICTEDLIMITS

We first prove that the auction is truthful. By Myerson's Lemma (Lemma 2.1), it suffices to prove the monotonicity of the allocation rule.

**Lemma C.1.** *For any $\epsilon \leq 2$, the* PREDICTEDLIMITS *auction is truthful.*

*Proof.* We show that the allocation rule of Auction 1 is monotone, and then invoke Myerson's lemma to conclude truthfulness.

Fix a facility $\tilde{\ell} \in L$, and hold constant the true opening costs $\mathbf{c}_{-\tilde{\ell}}$ of all other facilities as well as all predictions $\hat{\mathbf{c}}$. It suffices to prove that if $\tilde{\ell}$ belongs to the winning set when it reports cost $a \geq 0$, then it still belongs to the winning set when it reports any lower cost $b \leq a$.

Recall that the auction uses the modified cost function

$$c'_\ell(S) = \begin{cases} \frac{2}{\epsilon} c_\ell, & \text{if } S = \hat{\mathtt{OPT}} \text{ and } c_\ell > \hat{c}_\ell, \\ c_\ell, & \text{otherwise.} \end{cases}$$

We slightly abuse notation and use $o'_{\tilde{\ell}}(S, a)$ and $o'_{\tilde{\ell}}(S, b)$ to denote the scaled cost when facility $\tilde{\ell}$ reports $a$ and $b$, respectively. Since lowering $\tilde{\ell}$'s reported cost can only (weakly) decrease its scaled cost in *every* candidate set, and since the event "$S = \hat{\mathtt{OPT}}$" depends only on the prediction, not on the reported bid, we have

$$c'_{\tilde{\ell}}(S, b) \ \leq \ c'_{\tilde{\ell}}(S, a) \quad \text{for all } S.$$

Let $S^*$ be the winning set when $\tilde{\ell}$ bids $a$, and let $S'$ be the winning set when it instead bids $b$, keeping everything else fixed. If $S' = S^*$ then $\tilde{\ell} \in S'$ immediately. Otherwise the only cost that changed is $c'_{\tilde{\ell}}$, so in order for the auction to switch to a different optimal set $S'$, it must still include $\tilde{\ell}$. Hence in all cases $\tilde{\ell}$ remains selected, proving monotonicity. Finally by Myerson's lemma, a monotone allocation rule induces truthful payments, so the auction is truthful. $\square$

## C.2 Missing Proofs from Section 4

### C.2.1 Proof of Lemma 4.3

*Proof.* By definition of the auction, at most one subset, $\hat{\text{OPT}}$, can have its costs scaled. Therefore, it suffices to consider the relationship between $\hat{\text{OPT}}$ and $\text{OPT}$ (the true optimal solution). We also note that since $\epsilon \leq 2$, any set that is scaled can only increase its total cost relative to other solutions. The proof structure closely follows the VCG analysis. The main idea is to handle the two cases depending on whether the predicted optimal set $\hat{\text{OPT}}$ matches the true optimal set $\text{OPT}$.

**Case one: $\hat{\text{OPT}} \neq \text{OPT}$.** In this case, the true optimum $\text{OPT}$ was not the set subject to potential scaling. Since $S^* = \text{OPT}$ is the output, $\text{OPT}$ must remain the optimal solution even in the modified (potentially scaled) instance. We apply the same VCG analysis structure as in Theorem 3.1 and Lemma 4.2.

Define $\text{OPT}_f$ and $\pi_f(\ell)$ as in Theorem 3.1. The bounds from (2) and (3) hold, so we get

$$\sum_{\ell \in \text{OPT}} \left[ p_\ell + \sum_{f:\ell \in \text{OPT}_f} d(U_{\ell,f}, \ell) \right] \leq \sum_{f \in F} o'_f(F) + 2 \sum_{f \in F} \sum_{\ell \in \text{OPT}_f} d(U_{\ell,f}, f) + \sum_{\ell \in \text{OPT}} \sum_{f:\ell \in \text{OPT}_f} d(U_{\ell,f}, \ell)$$

$$\leq \frac{2}{\epsilon} \sum_{\ell \in F} o_\ell + 2d(U, F) + \sum_{\ell \in \text{OPT}} \sum_{f:\ell \in \text{OPT}_f} d(U_{\ell,f}, \ell)$$

$$\leq \frac{2}{\epsilon} \sum_{\ell \in F} o_\ell + 2d(U, F) + \left( \sum_{\ell \in F} o_\ell + d(U, F) \right)$$

$$= \left( \frac{2}{\epsilon} + 1 \right) \sum_{\ell \in F} o_\ell + 3 \, d(U, F), \tag{5}$$

where the second inequality assumes the worst case for $o'_f(F)$, which occurs if $F = \hat{\text{OPT}}$ and the opening costs of facilities in $F$ were under-predicted. The third inequality uses the fact that the connection cost of the true optimal solution is weakly less than the total cost of the frugal solution.

**Case two: $\hat{\text{OPT}} = \text{OPT}$.** Since $S^* = \text{OPT}$, the consistency analysis from Lemma 4.2 applies directly, and we have:

$$\sum_{\ell \in \text{OPT}} p_\ell + \sum_{\ell \in \text{OPT}} \sum_{f:\ell \in \text{OPT}_f} d(U_{\ell,f}, \ell) \leq (1+\epsilon) \cdot c(F) = (1+\epsilon) \left[ \sum_{\ell \in F} o_\ell + d(U, F) \right].$$

The lemma statement follows by taking the maximum of the bounds from the two cases $\qquad \square$

### C.2.2 Proof of Lemma 4.4

*Proof.* The proof is very similar to the VCG analysis, replacing the facilities in the optimal solution with the facilities in $S = S^* \setminus F$. In addition, we also need to consider the case where the corresponding frugal facility $f$ is included in $S^*$, which turns out to be an easier case for bounding rerouting costs.

Consider the case where $S^* \neq \text{OPT}$. First, we note that $\hat{\text{OPT}} = \text{OPT}$, otherwise $\text{OPT}$ would be output since scaling a solution only makes it worse. Therefore, both $S^*$ and $F$ are unscaled. Let $S = S^* \setminus F$.

Define $U_\ell$, $U_{\ell,f}$, $x_f(\ell)$ and $\pi_f(\ell)$ as in Theorem 3.1, but replacing $\text{OPT}$ with $S$. If $f \in S^*$, then rerouting users from $\ell \in S$ to $f$ is bounded by just connecting the users to $f$. Therefore, in the worst case, all facilities $f \in F$ are not in $S^*$. Then, inequalities (2) and (3) hold when replacing $\text{OPT}$ with $S$. Therefore, our total payment is bounded by

$$\sum_{\ell \in S} \left( p_\ell + \sum_{u \in U_\ell} d(u, \ell) \right) \leq \left( \sum_{f \in F} o_f + 2 \sum_{f \in F} \sum_{\ell \in S_f} d(U_{\ell,f}, f) \right) + \sum_{\ell \in S} \sum_{f:\ell \in S_f} d(U_{\ell,f}, \ell)$$

$$= \sum_{f \in F} [2d(U_j, f) + o_f] + \sum_{\ell \in S} \sum_{f:\ell \in S_f} d(U_{\ell,f}, \ell).$$

Cancelling the total connection cost term $\sum_{\ell \in S} \sum_{u \in U_\ell} d(u, \ell) = \sum_{\ell \in S} \sum_{f:\ell \in S_f} d(U_{\ell,f}, \ell)$ from both sides:

$$\sum_{\ell \in S} p_\ell \leq \sum_{f \in F} [2d(U_j, f) + o_f] \leq \sum_{\ell \in F} o_\ell + 2d(U, F). \qquad \Box.$$

### C.2.3 Proof of Lemma 4.6

*Proof.* Consider any instance $(U, \mathbf{o}, d)$ and any $\epsilon \leq 2$. Let OPT and $F$ be the optimal solution and frugal solution of the given instance. Let $S^*$ be the output of Auction 1.

**Case one:** $S^* = $ OPT. If $S^* = $ OPT, by Lemma 4.3 we have:

$$\sum_{\ell \in S^*} p_\ell + d(U, S^*) \leq \left( \frac{2}{\epsilon} + 1 \right) \sum_{\ell \in F} o_\ell + 3d(U, F).$$

**Case two:** $S^* \neq $ OPT. We now consider the case where $S^* \neq $ OPT. First, note that the *connection cost* of $S^*$ is weakly less than or equal to the total cost of any solution $K$, including the frugal solution $F$. We have:

$$d(U, S^*) \leq \sum_{\ell \in F} o'_\ell(F) + d(U, F) = \sum_{\ell \in F} o_\ell + d(U, F), \qquad (6)$$

where the equality is due to the fact that OPT is the solution that is scaled, by the fact that $S^* \neq $ OPT.

Let $S = S^* \setminus F$ and $S^f = S^* \cap F$. Putting together Lemma 4.4 and Lemma 4.5, we get:

$$
\begin{aligned}
\sum_{\ell \in S^*} p_\ell + d(U, S^*) &\leq \sum_{\ell \in S} p_\ell + \sum_{\ell \in S^f} p_\ell + d(U, S^*) \\
&\leq \sum_{\ell \in F} o_\ell + 2d(U, F) + \sum_{\ell \in S^f} p_\ell + d(U, S^*) \qquad \text{(by Lem. 4.4)} \\
&\leq \max\left( 3, \frac{2}{\epsilon} + 1 \right) \sum_{\ell \in F} o_\ell + \max\left( 4, \frac{2}{\epsilon} + 2 \right) d(U, F) + d(U, S^*) \\
&\qquad\qquad\qquad\qquad\qquad\qquad\qquad\qquad\qquad \text{(by Lem. 4.5)} \\
&\leq \max\left( 4, \frac{2}{\epsilon} + 2 \right) \sum_{\ell \in F} o_\ell + \max\left( 5, \frac{2}{\epsilon} + 3 \right) d(U, F) \qquad \text{(by (6))}
\end{aligned}
$$

Taking the worst case of the two scenarios we get that

$$\sum_{\ell \in S^*} p_\ell + d(U, S^*) \leq \max\left( 5, \frac{2}{\epsilon} + 3 \right) \left[ \sum_{i \in F} o_i + d(U, F). \right] \qquad \Box$$

## D Proof of Theorem 4.1

*Proof of Theorem 4.1.* Auction PREDICTEDLIMITS is truthful by Lemma C.1, $(1 + \epsilon)$-consistent by Lemma 4.2, and $\max\left( 5, \frac{2}{\epsilon} + 3 \right)$-robust by Lemma 4.6. $\qquad \Box$

## E Missing Proofs from Section 5

### E.1 Truthfulness of ERRORTOLERANT

We first prove the truthfulness of Auction ERRORTOLERANT. The proof is very similar to Theorem C.1.

**Lemma E.1.** *For any $\epsilon \leq 2$, the* PREDICTEDLIMITS *auction is Truthful.*

*Proof.* We show that the allocation rule of Auction 2 is monotone.

Fix a facility $\tilde{\ell} \in L$, and $\mathbf{o}_{-\tilde{\ell}}$ of all other facilities as well as all predictions $\hat{\mathbf{o}}$. Let $\tilde{\ell}$'s reported cost be $a \geq 0$, and suppose that when it bids $a$ the auction selects $S_a^* \ni \tilde{\ell}$. We must show that if it instead reports any lower cost $b \leq a$, it remains selected.

Denote by $c_\lambda(S\,;\,x)$ the total cost of set $S$ under report $x \in \{a, b\}$, according to Auction 2. Note that in either branch (the "whole-solution downscaling" or the "per-facility inflation"), lowering $\tilde{\ell}$'s cost from $a$ to $b$ *weakly decreases* its contribution to $c_\lambda(S)$ for *every* $S$. We consider two cases:

**Case 1: Whole-solution downscaling applies at report $a$.** That is,

$$\forall\,\ell' \in \hat{\mathsf{OPT}},\; o_{\ell'} \leq \lambda\,\hat{o}_{\ell'} \quad \text{under } x = a.$$

Since $b \leq a$, the same test holds at $x = b$, so the auction remains in the downscaling branch. In that branch

$$c_\lambda(S\,;\,x) = \begin{cases} \frac{1}{\lambda^2}\Big(d(U, S) + \sum_{\ell \in S} o_\ell\Big), & S = \hat{\mathsf{OPT}}, \\ d(U, S) + \sum_{\ell \in S} o_\ell, & \text{otherwise}, \end{cases}$$

and reducing $\tilde{\ell}$'s cost from $a$ to $b$ multiplies its term by at most $b/a \leq 1$ in every candidate set. Hence

$$c_\lambda(S\,;\,b) \;\leq\; c_\lambda(S\,;\,a) \quad \text{for all } S,$$

Since solution that got a reduced cost must contain $\tilde{\ell}$, any new minimizer $S_b^*$ must still contain $\tilde{\ell}$.

**Case 2: Per-facility inflation applies at report $a$.** Then there is some $\ell' \in \hat{\mathsf{OPT}}$ with $o_{\ell'} > \lambda\hat{o}_{\ell'}$ under $x = a$. Two subcases arise:

- If $\tilde{\ell} \notin \hat{\mathsf{OPT}}$. Lowering its bid does not affect the branch test, so we stay in the inflation branch. But in that branch each facility's scaled cost (whether inflated by $2/\epsilon$ or not) is a nondecreasing function of its report, so the argument from the original auction applies verbatim to show monotonicity.

- If $\tilde{\ell} \in \hat{\mathsf{OPT}}$. Then $a > \lambda\hat{o}_{\tilde{\ell}}$ but $b \leq \lambda\hat{o}_{\tilde{\ell}}$, so at $x = b$ the auction flips into the downscaling branch. In that branch it will choose $\hat{\mathsf{OPT}}$ (since we uniformly downscale its cost), and $\hat{\mathsf{OPT}} \ni \tilde{\ell}$. Thus $\tilde{\ell}$ remains selected.

In all cases, lowering $\tilde{\ell}$'s bid can only weakly decrease its cost in every candidate set including it, and any switch of the minimizer continues to include $\tilde{\ell}$. Therefore the allocation rule is monotone, and by Myerson's lemma the resulting payments make the auction truthful. $\qquad\square$

## E.2 Performance analysis for ERRORTOLERANT

We now provide the performance as a function of the error $\eta$ and the error tolerance parameter $\lambda$.

### E.2.1 Performance when $\eta \leq \lambda$

We first consider the case where $\eta \leq \lambda$, we start by showing the claim that $\hat{\mathsf{OPT}}$ is always outputted when $\eta \leq \lambda$. The result of this case is summarized in Lemma E.8. We first show that under the condition $\eta \leq \lambda$, the prediction optimal is always returned.

**Lemma E.2.** *If $\eta \leq \lambda$, then Auction 2 always selects $\hat{\mathsf{OPT}}$; that is, $S^* = \hat{\mathsf{OPT}}$.*

We now provide an upper bound of the connection cost of the output solution.

*Proof.* Let

$$\hat{c}(S) \;=\; d(U, S) \;+\; \sum_{\ell \in S} \hat{o}_\ell,$$

be the predicted total cost. First note that since $\eta \leq \lambda$, we have $o_\ell \leq \lambda\hat{o}_\ell$ for all $\ell$ the auction will set

$$c_\lambda(S) \;=\; \begin{cases} \dfrac{1}{\lambda^2}\,c(\hat{\mathsf{OPT}}), & S = \hat{\mathsf{OPT}}, \\ c(S), & S \neq \hat{\mathsf{OPT}}. \end{cases}$$

Since $\hat{\mathsf{OPT}}$ minimizes the *predicted* cost,

$$\hat{c}(\hat{\mathsf{OPT}}) \;\leq\; \hat{c}(S) \quad \forall\, S \subseteq L. \tag{7}$$

By $\eta \leq \lambda$ we have $o_\ell \leq \lambda \hat{o}_\ell$ for all $\ell \in \hat{\text{OPT}}$, so

$$c(\hat{\text{OPT}}) = \sum_u d(u, \hat{\text{OPT}}) + \sum_{\ell \in \hat{\text{OPT}}} o_\ell \leq \sum_u d(u, \hat{\text{OPT}}) + \lambda \sum_{\ell \in \hat{\text{OPT}}} \hat{o}_\ell \leq \lambda \hat{c}(\hat{\text{OPT}}),$$

On the other hand, since $\hat{o}_\ell \leq \eta \, o_\ell$ for all $\ell$, we get

$$\hat{c}(S) = \sum_u d(u, S) + \sum_{\ell \in S} \hat{o}_\ell \leq \sum_u d(u, S) + \eta \sum_{\ell \in S} o_\ell = \eta \, c(S),$$

Combining the above inequality with (7) we have,

$$c(\hat{\text{OPT}}) \leq \lambda \hat{c}(S) \leq \lambda \eta \, c(S) \implies \frac{1}{\lambda^2} c(\hat{\text{OPT}}) \leq c(S) \implies c_\lambda(\hat{\text{OPT}}) \leq c_\lambda(S), \quad \forall S \neq \hat{\text{OPT}}.$$

Since the auction output the set that minimizr $c_\lambda$, we therefore have $S^* = \hat{\text{OPT}}$. $\qquad \square$

**Observation E.3.** *For any set $S \neq \hat{\text{OPT}}$, let $\hat{c}(S) = d(U, S) + \sum_{\ell \in S} \hat{o}_\ell$, we have*

$$\hat{c}(S) \leq \eta c(S).$$

*Proof.* $\eta \leq \lambda$ implies $\hat{o}_\ell \leq \lambda \, o_\ell$ for every $\ell$, so

$$\hat{c}(S) = d(U, S) + \sum_{\ell \in S} \hat{o}_\ell \leq d(U, S) + \eta \sum_{\ell \in S} o_\ell = \eta \Big( d(U, S) + \sum_{\ell \in S} o_\ell \Big). \qquad \square$$

We now provide an upper bound of the connection cost of the output solution.

**Lemma E.4.** *If $\eta \leq \lambda$, then for any $S \subseteq L$,*

$$d\left(U, \hat{\text{OPT}}\right) \leq \eta \left( d(U, S) + \sum_{\ell \in S} o_\ell \right).$$

*Proof.* Since $d\left(U, \hat{\text{OPT}}\right) \leq \hat{c}(\hat{\text{OPT}})$, combining with the fact that $\hat{\text{OPT}}$ minimizes $\hat{c}$, we have gives

$$d\left(U, \hat{\text{OPT}}\right) \leq \hat{c}(\hat{\text{OPT}}) \leq \hat{c}(S) \leq \eta \Big( d(U, S) + \sum_{\ell \in S} o_\ell \Big),$$

where the last inequality is by Observation E.3. $\qquad \square$

We now move on to upper bound the payment to the facilities in the output solution, i.e., $\hat{\text{OPT}}$. In particular, we will partition $\hat{\text{OPT}}$ based on the threshold they applied, and whether it is a member of $F$, the true frugal solution. We first analyze the low-threshold facilities.

**Lemma E.5.** *Under the condition $\eta \leq \lambda$, let $S^1 = \left\{ \ell \in \hat{\text{OPT}} : p_\ell \leq \lambda \hat{o}_\ell \right\}$. Then*

$$\sum_{\ell \in S^1} p_\ell \leq \lambda \eta \, c(F),$$

*where $F$ is the frugal solution.*

*Proof.* Since $p_\ell \leq \lambda \hat{o}_\ell$ for every $\ell \in S^1$,

$$\sum_{\ell \in S^1} p_\ell \leq \lambda \sum_{\ell \in S^1} \hat{o}_\ell \leq \lambda \sum_{\ell \in \hat{\text{OPT}}} \hat{o}_\ell \leq \lambda \hat{c}(\hat{\text{OPT}}) \leq \lambda \hat{c}(F), \leq \lambda \eta \, c(F),$$

where the last inequality is by Observation E.3. $\qquad \square$

**Lemma E.6.** *Assume $\eta \leq \lambda$, and let*

$$S^2 = \left\{ \ell \in \hat{\text{OPT}} : p_\ell > \lambda \hat{o}_\ell, \ \ell \notin F \right\}.$$

*Then*

$$\sum_{\ell \in S^2} p_\ell \leq \epsilon \, c(F),$$

*where $F$ is the frugal solution.*

*Proof.* We follow the rerouting argument of Lemma 4.4, with the additional fact that when the reported bid is more than $\lambda \hat{o}_\ell$, the opening cost is scaled up. specialized to the output $\widehat{\text{OPT}}$.

Define $U_{\ell,f}$ and $\pi_f(\ell)$ as in Theorem 3.1, but replacing $\text{OPT}$ with $S^2$. If $f \in S^*$, then the cost of rerouting users from $\ell \in S^2$ to $f$ is bounded by just connecting the users to $f$. Therefore, in the worst case, all facilities $f \in F$ are not in $S^*$. Then, inequalities (2) and (3) hold when replacing $\text{OPT}$ with $S$. For $\ell \in S^2$, payments are scaled by $\frac{2}{\epsilon}$. Following a similar argument as Lemma 4.4, our total payment is bounded by

$$\frac{2}{\epsilon} \sum_{\ell \in S^2} p_\ell + \sum_{\ell \in S^2} \sum_{f:\ell \in S_f^2} d(U_{\ell,f}, \ell) \leq \sum_{f \in F} \left[ 2 \sum_{\ell \in S_f^2} d(U_{\ell,f}, f) + o_f + \sum_{\ell \in S_f^2} d(U_{\ell,f}, \ell) \right]$$

$$= \sum_{f \in F} \left[ 2\, d(U_f, f) + o_f \right] + \sum_{\ell \in S^2} \sum_{f:\ell \in S_f^2} d(U_{\ell,f}, \ell).$$

Cancelling the total connection cost term $\sum_{\ell \in S^2} \sum_{f:\ell \in S_f^2} d(U_{\ell,f}, \ell)$ from both sides gives

$$\frac{2}{\epsilon} \sum_{\ell \in S^2} p_\ell \leq \sum_{f \in F} \left[ 2\, d(U_f, f) + o_f \right],$$

we therefore get

$$\sum_{\ell \in S^2} p_\ell \leq \frac{\epsilon}{2} \sum_{f \in F} o_f + \epsilon \sum_{f \in F} d(U_f, f) \leq \epsilon \cdot c(F). \qquad \square$$

**Lemma E.7.** *Under the condition $\eta \leq \lambda$, let*

$$S^3 = \left\{ \ell \in \widehat{\text{OPT}} : p_\ell > \lambda \hat{o}_\ell,\ \ell \in F \right\}.$$

*Then*

$$\sum_{\ell \in S^3} p_\ell \leq \epsilon \cdot c(\text{OPT}).$$

*Proof.* For $\ell \in S^3$, $\ell \in F$ and $p_\ell > \lambda \hat{o}_\ell$. An argument analogous to Lemma 4.5 shows that inflating the opening cost of any facility in the frugal solution beyond its predicted cost by up to a $2/\epsilon$ factor bounds its threshold payment by $\epsilon$ times the cost of the true optimum. Hence $\sum_{\ell \in S^3} p_\ell \leq \epsilon\, c(\text{OPT})$. $\qquad \square$

**Lemma E.8.** *Assume $\eta \leq \lambda$. Let $F$ be the frugal solution, and let $S^*$ and $\{p_\ell\}_{\ell \in S^*}$ be the output of Auction 2. Then*

$$d(U, S^*) + \sum_{\ell \in S^*} p_\ell \leq \big(\eta(1+\lambda) + 2\epsilon\big) \cdot c(F).$$

*Proof.* By Lemma E.2, under $\eta \leq \lambda$ the auction selects $S^* = \widehat{\text{OPT}}$. We partition the payments over $\widehat{\text{OPT}}$ into three parts $S^1, S^2, S^3$ as in LemmasE.5, Lemma E.6 and Lemma E.7. since $c(\text{OPT}) \leq c(F)$. Summing connection cost and all payments gives

$$d(U, S^*) + \sum_{\ell \in S^*} p_\ell = d(U, \widehat{\text{OPT}}) + \sum_{\ell \in S^1} p_\ell + \sum_{\ell \in S^2} p_\ell + \sum_{\ell \in S^3} p_\ell$$

$$\leq \eta\, c(F) + \sum_{\ell \in S^1} p_\ell + \sum_{\ell \in S^2} p_\ell + \sum_{\ell \in S^3} p_\ell \qquad \text{(by Lem. E.4)}$$

$$\leq \eta\, c(F) + \lambda\eta\, c(F) + \sum_{\ell \in S^2} p_\ell + \sum_{\ell \in S^3} p_\ell \qquad \text{(by Lem. E.5)}$$

$$\leq \eta\, c(F) + \lambda\eta\, c(F) + \epsilon\, c(F) + \sum_{\ell \in S^3} p_\ell \qquad \text{(by Lem. E.6)}$$

$$\leq \eta\, c(F) + \lambda\eta\, c(F) + \epsilon\, c(F) + \epsilon\, c(F) \qquad \text{(by Lem. E.7)}$$

$$= \big(\eta(1+\lambda) + 2\epsilon\big) \cdot c(F). \qquad \square$$

### E.2.2 Performance for general error $\eta$

We now analyze performance for arbitrary $\eta$ by breaking the proof into two dimensions: (a) which scaling rule the auction applies: either whole-solution downscaling or per-facility inflation—and (b) which set is ultimately selected. In each of the resulting subcases, we separately bound the connection cost and the total payment using rerouting arguments and the appropriate scaling factor; combining these four analyses yields the unified frugality guarantee stated in Lemma E.11. We start by consider the case where *whole-solution scaling* is applied and the $\hat{\mathtt{OPT}}$ is outputed.

**Lemma E.9.** *Let $S^*$ be the output of Auction 2, if $S^* = \hat{\mathtt{OPT}}$, for any $S \neq \hat{\mathtt{OPT}}$,*

$$d\left(U, S^*\right) \;\leq\; \lambda^2\, c(S).$$

*Proof.* By the scaling scehem of the auction we have $d\left(U, \hat{\mathtt{OPT}}\right) \leq \lambda^2 c_\lambda(\hat{\mathtt{OPT}})$, combining with the fact that $S^*$ minimizes $c_\lambda$, we have gives

$$d\left(U, S^*\right) \;\leq\; \lambda^2 \cdot c_\lambda(S^*) \;\leq\; \lambda^2 \cdot c_\lambda(S) \;\leq\; \lambda^2 \cdot c(S)$$

where the last inequality is since $S \neq \hat{\mathtt{OPT}}$. $\qquad\square$

**Lemma E.10.** *Let $S^*$ be the output of Auction 2. If $S^* = \hat{\mathtt{OPT}}$ and the whole-solution-scaling is applied, then*

$$\sum_{\ell \in S^*} p_\ell \;\leq\; (2\lambda^4 + 2\lambda^2)\, c(F),$$

*Proof.* We follow the rerouting argument of Lemma 4.4. Partition $\hat{\mathtt{OPT}}$ into

$$S \;=\; S^* \setminus F \quad \text{and} \quad S^f \;=\; \hat{\mathtt{OPT}} \cap F.$$

Define $U_{\ell,f}$ and $\pi_f(\ell)$ as in Theorem 3.1 but replace OPT with $S^*$

Since the auction applies a $1/\lambda^2$ down-scaling to $\hat{\mathtt{OPT}}$, no profitable deviation implies

$$\frac{1}{\lambda^2}\left(p_i + \sum_{j:\ell \in S_f} d(U_{\ell,f}, \ell)\right) \;\leq\; \sum_{j:\ell \in S_f} r_{\ell,f}, \tag{8}$$

for each $\ell \in S$, where $r_{\ell,f}$ is the cost to reroute $U_{\ell,f}$ to $\pi_f(\ell)$.

By the standard charging argument,

$$\sum_{\ell \in S_f} r_{\ell,f} \;\leq\; 2\sum_{\ell \in S_f} d(U_{\ell,f}, f) + o_f + \sum_{\ell \in S_f} d(U_{\ell,f}, \ell). \tag{9}$$

Summing (8) over all $\ell \in S$ and then using (9) together with Lemma E.9 ($\sum_f d(U_f, f) \leq d(U, F)$) yields

$$\frac{1}{\lambda^2}\sum_{\ell \in S} p_i \;\leq\; 2\,c(F) + \left(1 - \tfrac{1}{\lambda^2}\right)\sum_{\ell \in S}\sum_{j:\ell \in S_f} d(U_{\ell,f}, \ell) \;\leq\; 2c(F) + (\lambda^2 - 1)\,c(F).$$

Hence

$$\sum_{\ell \in S} p_\ell \;\leq\; (\lambda^4 + \lambda^2)\, c(F).$$

The case $f \in S^*$ gives the even better bound $r_{\ell,f} \leq d(U_{\ell,f}, f)$, so it cannot increase the total.

Applying the identical rerouting argument to $S^f$ (with $F$ replaced by OPT) yields the same upper bound $(\lambda^4 + \lambda^2)c(F)$. Summing over both parts,

$$\sum_{\ell \in S^*} p_\ell = \sum_{\ell \in S} p_\ell + \sum_{\ell \in S^f} p_\ell \;\leq\; (2\lambda^4 + 2\lambda^2)\, c(F). \qquad\square$$

**Lemma E.11.** *The Auction 2 achieves a frugality ratio*

$$\max\left\{ 2\lambda^4 + 3\lambda^2,\; 3 + \tfrac{2}{\epsilon}\right\}.$$

*Proof.* The auction uses exactly one of two branches:

**(i) Whole-solution downscaling:**

$$C_\lambda(S) = \begin{cases} \frac{1}{\lambda^2}\left(d(U,S) + \sum_{\ell \in S} o_\ell\right), & S = \hat{\mathtt{OPT}}, \\ d(U,S) + \sum_{\ell \in S} o_\ell, & S \neq \hat{\mathtt{OPT}}. \end{cases}$$

**(ii) Per-facility inflation:**

$$C_\lambda(S) = d(U,S) + \sum_{\ell \in S} o'_\ell(S),$$

where $o'_\ell(S) = \frac{2}{\epsilon} o_\ell$ if $S = \hat{\mathtt{OPT}}$ and $o_\ell > \lambda \hat{o}_\ell$, and $o'_\ell(S) = o_\ell$ otherwise.

**Case 1: Whole-solution downscaling applies.**

Case 1.1: If the auction still outputs $S^* = \hat{\mathtt{OPT}}$, then by Lemma E.9,

$$d(U, S^*) \leq \lambda^2 \, c(F).$$

Combined with Lemma E.10, which gives $\sum_{\ell \in S^*} p_\ell \leq (2\lambda^4 + 2\lambda^2) \, c(F)$, we obtain

$$d(U, S^*) + \sum_{\ell \in S^*} p_\ell \leq (2\lambda^4 + 3\lambda^2) \, c(F).$$

Case 1.2: Otherwise $S^* \neq \hat{\mathtt{OPT}}$. In this subcase, $\hat{\mathtt{OPT}}$ was downscaled but not chosen, and all other candidate sets remain unscaled. Thus from the winners' perspective the outcome is no worse than standard VCG, so

$$d(U, S^*) + \sum_{\ell \in S^*} p_\ell \leq 3 \, c(F).$$

**Case 2: Per-facility inflation applies.** In this branch the analysis of Section 4.3 carries over unchanged: the exact triggering condition ($o_\ell > \hat{o}_\ell$ vs. $o_\ell > \lambda \hat{o}_\ell$) does not affect the worst-case bound. By the robustness proof of Theorem 4.1,

$$d(U, S^*) + \sum_{\ell \in S^*} p_\ell \leq \max\left\{5, \, 3 + \tfrac{2}{\epsilon}\right\} c(F).$$

Taking the maximum over all branches and subcases yields

$$\max\left\{ 2\lambda^4 + 3\lambda^2, \, 5, \, 3 + \tfrac{2}{\epsilon}\right\} = \max\left\{2\lambda^4 + 3\lambda^2, \, 3 + \tfrac{2}{\epsilon}\right\} c(F). \qquad \square$$

Putting the findings together gives up the performance of ERRORTOLERANT.

*Proof of Theorem 5.1.* If $\eta \leq \lambda$, by Lemma E.8 have the first statement and Lemma E.11 proves the other bound, for arbitrary $\eta$ values. $\qquad \square$

## F    Computational Complexity

The standard IP formulation for UFL uses binary variables $x_\ell \in \{0,1\}$ to denote whether facility $\ell \in L$ is opened and variables $y_{u\ell} \in [0,1]$ for assigning user $u \in U$ to facility $\ell$.

The implementation of `Auction 1` proceeds as follows:

1. **Determine the Predicted Optimal Set:** First, we solve a classic UFL integer program using the *predicted* opening costs, $\{\hat{o}_\ell\}_{\ell \in L}$, to find the predicted optimal set, $\hat{OPT}$.

2. **Find the Best Alternative Solution:** After identifying $\hat{OPT}$, its modified cost according to the auction rules is calculated directly. To find the best alternative, we solve a second UFL

integer program using the *true* opening costs, $\{o_\ell\}_{\ell \in L}$. A linear constraint is added to this IP to make the set $O\hat{P}T$ an infeasible solution:

$$\sum_{\ell \in O\hat{P}T} x_\ell \leq |O\hat{P}T| - 1$$

This inequality ensures that at least one facility from the set $O\hat{P}T$ remains unopened, allowing the IP to find the optimal solution among all other feasible sets.

3. **Select the Winning Set:** The auction's final output, $S^*$, is determined by comparing the modified cost of $O\hat{P}T$ with the cost of the best alternative solution found in the previous step. The set with the lower cost is chosen.

4. **Compute Payments:** Finally, for each facility $\ell$ in the winning set $S^*$, its threshold payment is computed. This is done by solving one additional UFL integer program per winning facility to find the critical cost at which that facility would no longer be part of the optimal solution.

