# OpenReview forum: "Procurement Auctions with Predictions: Improved Frugality for Facility Location"
_NeurIPS.cc/2025/Conference — NeurIPS 2025 poster_

### Official Review · Reviewer_cHUk · 2025-06-15

**Clarity:** 3
**Significance:** 3
**Originality:** 2
**Rating:** 5
**Confidence:** 3

**Summary:**

This work examines a variant of the learning-augmented facility location auction problem, where each facility must be explicitly rented or purchased to be utilized. This introduces incentives not for the facility users—as in prior work—but also for the facility owners. The main performance metric studied is the *frugality ratio*.

**Key results**:

- Without predictions: The classic VCG auction is shown to achieve a frugality ratio of exactly **3**.

- With predictions:
  1. An auction that guarantees a $(1+\varepsilon, \max\{5, \frac{2}{\varepsilon}\})$ consistency-robustness trade-off.
  2. A smoother alternative trade-off is also given:
     - If $(\eta \le \lambda)$, the frugality is bounded by $\eta(1+\lambda) + 2\varepsilon$.
     - Otherwise, the bound becomes $\max $ {$\{ 2\lambda^4 + 3\lambda^2, \ 3 + \frac{2}{\varepsilon} \}$ }.

     where $\varepsilon \in (0,2]$ and $\lambda > 1$ are tunable parameters.

**Questions:**

* Are there negative results about strategic UFL for the approximation ratio objective? The paper would benefit from listing impossibilities or hardness result of achieving a constant approximation ratio (with and without predictions).


* In line 169 $U_{l,f}$ is used even though it was not defined anywhere. From the context, it seems like $U_{l,f}$ is defined to be the set of users that are connected to both facilities $l \in OPT$ in the optimal solution and to $f \in F$ in the frugal solution. Is this true?

* You have shown a lower bound of $3$ for the tree metric for the frugal ratio. Are you aware if the results are different for a Euclidean metric space (the real line, the plane, $R^d$ Eucledian space)?

* In Appendix B: line 457 states "Figure B" but the caption says "Figure 2". Please choose one of these (probably 2). Also in the same line, I think the edges should be $(l_i,u_i)$ instead of $(l,u_i)$. Line 458: "distance of from" - remove the "of". Line 459: $d(l,l_j)$ should be $d(l_i, l_j)$. The same goes for the end of the sentence.

* Putting the ErrorTolerant auction in the paper body rather than the appendix (in the version where there are $\ge$ 10 pages to the paper body rather than $9$) might be a good idea.

* What happens if there's one erroneous facility cost prediction (say one predicted cost is far from the actual cost) but all the rest are not? Does the ErrorTolerant auction get any guarantee? It is fine if it is unknown to you.

**Ethical Concerns:**

["NO or VERY MINOR ethics concerns only"]

**Final Justification:**

In the rebuttal the authors convinced me their analysis techniques are different enough even though the VCG mechanism is used. Together with a few other smaller points addressed, I've decided to raise my score.

The main concern remaining is that it seems hard to choose reasonable $\lambda,\varepsilon$ parameters that will make the given auction with predictions more attractive than the no-predictions one.

Reading the other reviews I was not convinced there is an ethical concern. In terms of fairness, this is a different metric that is studied in algorithmic game theory, and often fairness and other metrics (such as social welfare or revenue) do not go hand in hand and it is ok. Some work consider obtaining both at the same time, but this is not the focus of this work.

**Limitations:**

yes

**Paper Formatting Concerns:**

No formatting concerns.

**Quality:**

3

**Strengths And Weaknesses:**

Strengths:

* The VCG no-predictions frugal analysis is simple (elegant), yet tight.

* The paper achieves  consistency-robustness results for the settings with predictions. The Error Tolerant Scaled VCG auction allows smooth consistency-robust results with tunable parameters $\lambda$, $\varepsilon$.

Weaknesses:

* The problem is not about strategic facility location as much as it is about auctions. That is, the users are not strategic and all of their locations are known (the user locations only affect the cost of each solution). This reduces to an auction where the auction designer attempts to buy/rent the facility locations from the owners. Hence, the standard techniques of VCG and Myersons lemma apply. In terms of the combination with predictions, the techniques seem similar to the ones given for auctions with predictions by Xu and Lu 2022 - the $\lambda$ parameter seems (roughly) to play the role of the $\gamma$ parameter of Xu and Lu 2022 (their Theorem 2,5).

* The PredictedLimits auction is really only useful if the optimal solution is exactly correct, which seems unrealistic (a problem which the "ErrorTolerant" auction attempts to solve).

* While the last ("ErrorTolerant") auction is more robust to small errors than the PredictedLimits one, its performance of the Error Tolerant auction only seems to work well for very good predictions. $\lambda$ seems to roughly be the user's estimate of $\eta$. Say the auctioneer chooses a reasonable estimate - that the worst of all predictions is off by 50 percent ($\lambda = 1.5$). And say that in reality $\eta = 1.2$ (the worse prediction is off by 20 percent). Then the algorithm will get a frugality of $3$ which is what the VCG auction without predictions achieves. On the other hand, the robustness guarantees are not really small either ($16.875$ for $\lambda = 1.5$).
Note that also the $\varepsilon$ parameter must be small enough to allow the $\eta(1+\lambda)+2\varepsilon$ term be smaller than $0.5$ to allow any better than $3$ result in the best-case, but this results in a large robustness guarantee. Say for example, the user chooses reasonable $\varepsilon = \frac{1}{3}$ and $\lambda = 1.15$. So the robustness is $9$ (high), but, the consistency is around $3$ for reasonable error of $\eta = 1.1$, which is again what we would get with the no-predictions regime.
**To summarize this point**: it seems hard to choose reasonable $\lambda,\varepsilon$ parameters that will make the given auction with predictions more attractive than the no-predictions one.

---

> ### Author Rebuttal · Authors · 2025-07-31
>
> Thank you for carefully reviewing our paper and providing constructive feedback. We apologize for omitting the definition of  $U_{\ell, f}$, which is the set of users assigned to $\ell$ in OPT and to $f$ in the frugal solution $F$. We will make sure to address this point, along with other minor issues in the next revision of the paper. We address your questions below:
>
> * **Regarding the techniques and comparison to Mechanism Design with Predictions literature**
>
> Personalized scaling is a widely used approach for leveraging predictions, as seen in [Xu and Lu IJCAI ‘22] and many others, including [Balkanski, Gkatzelis, Tan ITCS ‘23], [Berger et al. EC ‘24], [Balkanski et al. EC ‘24], and [Christodoulou, Sgouritsa, Vlachos NeurIPS ‘24]. The key distinction between our mechanism and all these scaling-based methods is that our scaling is solution-based rather than facility-based. Specifically, we only scale the facilities in $\hat{OPT}$ when they are part of the predicted optimal solution; all other facilities, as well as those in $\hat{OPT}$ when considered in other solutions, are not scaled. This design avoids unnecessary distortion from inaccurate predictions. We experimented with direct scaling approaches similar to prior work but could not achieve constant robustness. Furthermore, our setting differs from that of Xu and Lu, who, to our knowledge, are the only other procurement-focused work in this line: their objective is payment only, whereas ours also includes connection cost, which can change in complex, non-smooth ways as the solution changes. This makes performance analysis significantly more challenging in our setting.
>
> [Xu and Lu IJCAI ‘22]  Chenyang Xu, Pinyan Lu.  Mechanism Design with Predictions 2022
>
> [Balkanski, Gkatzelis, Tan ITCS ‘23] Eric Balkanski, Vasilis Gkatzelis, Xizhi Tan. Strategyproof Scheduling with Predictions 2023
>
> [Berger et al. EC ‘24] Ben Berger, Michal Feldman, Vasilis Gkatzelis, Xizhi Tan. Learning-Augmented Metric Distortion via (p,q)-Veto Core 2024
>
> [Balkanski et al. EC ‘24] Eric Balkanski, Vasilis Gkatzelis, Xizhi Tan, Cherlin Zhu. Online Mechanism Design with Predictions 2024
>
> [Christodoulou, Sgouritsa, Vlachos NeurIPS ‘24] George Christodoulou, Alkmini Sgouritsa, Ioannis Vlachos. Mechanism design augmented with output advice 2024
>
> * **Regarding lower bounds**
>
> Although we are not aware of any general lower bounds for the frugality ratio of all truthful mechanisms, we do actually have additional lower bounds for the VCG mechanisms in interesting classes of metric spaces which we did not include in the submission. Specifically, for the line metric, a similar construction that led to the lower bound of 3 for general metrics (with twists on the edge weights) yields a lower bound of 5/3 for the VCG auction (in fact, we can also show that this bound is tight for the line). Also, for $d$-dimensional Euclidean spaces with $d \ge 2$, a similar construction with optimized weights  provides a lower bound of $2d-\frac{(2d-1)(4d^{2}-1)}{4d^{2}-2d+1+2\sqrt{5d^{2}-2d}}$. This value is approximately 1.857 for $d=2$ and converges to $\sqrt{5} \approx 2.236$ as $d$ approaches infinity. We would be happy to include this results in the paper if the reviewer believes that they add significant value.
>
> * **What happens if there's one erroneous facility cost prediction (say one predicted cost is far from the actual cost) but all the rest are not? Does the ErrorTolerant auction get any guarantee?**
>
> Thank you for raising this interesting question! We considered this question and we can show that when only one facility cost is mispredicted, the ErrorTolerant auction actually achieves much better guarantees than the general bound. In particular, if $\eta$ is the ratio of the single misprediction, the improved frugality ratios are:
>
> * When $\eta < \lambda$: The ratio is at most **$\lambda^2 + \epsilon$**, and in most sub-cases, it improves to $\lambda + 2\epsilon$ or better.
> * When $\eta \ge \lambda$: The ratio is bounded by **$\max\{5, 3 + 2/\epsilon\}$**, matching the robustness guarantee of the simpler Auction 1.
>
> We provide a proof sketch for this additional result below.
>
> **Proof Sketch**
>
> Let the facility with the mispredicted cost be $f$.
>
> *Case 1:* $f$ is not in OPT and is overpredicted.
>
> Since $f$ was not in the true OPT and is now even less attractive, the predicted optimal set remains the true optimal set ($\hat{OPT} = OPT$). The whole-solution scaling rule applies, and OPT is selected. The analysis is much stronger than the general case because every facility in the chosen solution has a perfect prediction, which tightens the bounds from Lemmas D4, D5, and D6 to give a frugality ratio of $\lambda + \epsilon$.
>
> *Case 2:* $f$ is not in OPT and is underpredicted.
> * If $\hat{OPT} = OPT$: The analysis from Case 1 applies, yielding a $\lambda + \epsilon$ ratio.
> * If $\hat{OPT} \neq OPT$: This must be because the underpredicted cost made $f$ attractive enough to be included in $\hat{OPT}$.
>     - If $\eta \le \lambda$, the analysis from the paper gives a bound of $\lambda + 2\epsilon$.
>     - If $\eta \ge \lambda$, the per-facility scaling rule is triggered. As shown in Lemma D11, this bounds the frugality ratio by $\max\{5, 3 + 2/\epsilon\}$.
>
>
> *Case 3:* $f$ is in OPT and is overpredicted.
> * If $\hat{OPT} = OPT$: The whole-solution scaling is applied, and OPT is returned. A more careful analysis similar to Lemma D10, but accounting for only one misprediction, gives a frugal ratio of $\lambda^2 + \epsilon$.
> * If $\hat{OPT} \neq OPT$: This means f was priced out of the predicted solution. The $\hat{OPT}$ set contains only correctly predicted facilities. Whole-solution scaling applies to $\hat{OPT}$, and the resulting bound is $\lambda + 2\epsilon$.
>
> *Case 4:* $f$ is in OPT and is underpredicted.
> * If $\hat{OPT} = OPT$:
>     - If $\eta \le \lambda$, whole-solution scaling applies, and the analysis from Case 1 gives a $\lambda + \epsilon$ ratio.
>     - If $\eta \ge \lambda$, per-facility scaling is triggered, bounding the ratio by $\max\{5, 3 + 2/\epsilon\}$.
> * If $\hat{OPT} \neq OPT$: The analysis is identical to the second sub-case of Case 2.

---

> ### Comment · Reviewer_cHUk · 2025-08-02
>
> I thank the authors for their response.
>
> I trust that the minor issues—such as the definition and use of
> $U_{l,f}$ , which was raised by multiple reviewers—will be addressed in the revised version of the paper.
>
> * On the techniques and comparison to the Mechanism Design with Predictions literature:
>
> I agree that the performance analysis in your setting is more involved than in Xu and Lu 2022, and appreciate the clarification.
>
> * On lower bounds:
>
> Thank you for the detailed explanation. I found this aspect interesting and suggest briefly including the key insight (perhaps in 2–3 sentences) in the main text, with full details deferred to the appendix.
>
> * On robustness to a single erroneous prediction:
>
> I may not have phrased my question clearly. What I meant is: suppose it is promised that exactly one facility cost prediction may be arbitrarily incorrect (unbounded error), while all others are perfectly accurate. Under such an error model, can the ErrorTolerant auction still provide any meaningful performance guarantee?

---

> > ### Author Response · Authors · 2025-08-05
> >
> > Thank you for the clarification! If we now understand correctly, what the reviewer has in mind is an alternative error parameter: rather than quantifying the maximum ratio over all mispredictions, like our error parameter does, this alternative error parameter quantifies the number of mispredictions, without restricting the extent to which they are mispredicted.
> >
> > These two error parameters are clearly quite different, and the short answer is that our ErrorTolerant auction is optimized for the former and does not simultaneously guarantee improved results for the latter. Specifically, note that the ErrorTolerant auction takes as input two parameters, $\epsilon$ and $\lambda$. For the set of instances that the reviewer is suggesting (i.e., ones with a single, but unbounded, misprediction), the best choice would be to set $\epsilon$  to 0 and set $\lambda$ to 1, in which case the ErrorTolerant auction reduces to the VCG auction and yields a frugality ratio of 3.
> >
> > If we were to design an alternative auction optimized for the class of instances that the reviewer is suggesting (i.e., ones with a single, but unbounded, misprediction), we believe there is a much simpler solution that guarantees a frugality ratio of 1. Specifically, this auction just offers to each bidder a take-it-or-leave-it price, equal to their predicted cost. Then, it observes what subset of bidders accepted that price and it returns the solution that minimizes the total cost. The fact that it is truthful is straightforward, since the prices that each bidder is offered is only a function of the prediction, without any input from the bidders themselves. To verify that the frugality ratio is 1 for the particular subset of instances, we just need to consider two cases: whether the mispredicted facility is in the optimal solution, OPT, with respect to the true costs, or not. If not, then all the bidders from OPT accept the price they were offered and they win (leading to cost even less than the frugal solution). If, on the other hand, the mispredicted facility is in OPT, then the frugal solution (which is disjoint from OPT) is correctly predicted and all its bidders accept the prices that they were offered. As a result, this mechanism can achieve the frugal cost by accepting all the bidders from the frugal solution, and the only reason why the mechanism would choose an alternative solution is because it is even cheaper, implying the frugality ratio of 1.
> >
> > We would be happy to add these observations if the reviewer believes they would add value to the paper. Also, we could also add a discussion regarding an interesting recent paper titled "MAC Advice for Facility Location Mechanism Design" by Barak-Gupta-Talgam Cohen that introduces a new notion of "mostly" and "approximately" correct (or MAC for short) predictions under which the reviewer’s example would have small error. It would be an interesting direction for future work to consider both the alternative error parameter proposed by the reviewer and this very recent "mostly" and "approximately" correct error measure for this problem in future work.

---

> > > ### Comment · Reviewer_cHUk · 2025-08-09
> > >
> > > I thank the authors for their reply. The proposed future directions on alternative modeling of the error parameters are intriguing, and I agree that adding a brief discussion on this topic would add value to the paper.
> > >
> > > Final comments:
> > > My final concern from the weakness section of my review—regarding the parameter choice in ErrorTolerant—still stands. I encourage the authors to address this in the paper, perhaps in the conclusions section. Developing an improved mechanism in this regard would also be an interesting avenue for future work.
> > >
> > > I also suggest emphasizing the difference from previous work in terms of techniques.
> > >
> > > In light of all of the discussion above, I am raising my score.
> > >
> > > I have no further questions.

---

### Official Review · Reviewer_ENCf · 2025-06-27

**Clarity:** 4
**Significance:** 3
**Originality:** 4
**Rating:** 5
**Confidence:** 4

**Summary:**

The authors study a procurement version of the strategic facility location problem where facilities report their operating costs, and a central planner decides on which facilities to open to minimize operating costs plus the costs of serving a fixed set of users. The planner has access to predictions about the facilities' operating costs. The work contributes to a recent active line of study on mechanism design with predictions, so it is a timely contribution. The main performance metric is the frugality ratio, which compares a mechanisms cost to the cost of the second-best solution. The authors first improve a result on the frugality ratio of VCG (establishing a tight result), then present a mechanism attaining (1+eps)-consistency and max(5, 3+2/eps)-robustness, and then present a error-tolerant mechanism with consistency and robustness ratios depending on the accuracy of the cost predictions.

(Prior work on strategic facility location assumes strategic agents who can misreport their true locations. The present work considers agents with fixed locations, and the facilities are the strategic entities that need to report their operating costs.)

**Questions:**

What is the equivalent notion of frugality for a forward auction (1 seller, many buyers)? Why not just compare directly to OPT instead---that seems like a stronger benchmark and is what the literature on learning-augmented mechanism design (e.g. Xu and Lu [IJCAI'22], Balcan et al. [NeurIPS'23]) has typically focused on. My guess is that this is a feature specific to the facility-location/procurement setting, but some clarification would be helpful.

Related to the above, if one receives perfect predictions of the opening costs, isn't it possible to obtain cost = OPT while remaining truthful? (In contrast to the weaker result presented here that achieves a frugality ratio of 1.) Along that vein, consider the randomized mechanism that w.p. p trusts the predictions completely and implements OPT by paying each facility exactly the predicted cost, and w.p. 1-p discards the predictions and uses VCG. The consistency (I'm not sure about robustness) of this mechanism can be compared directly to OPT instead of the second-best. Where does this fit into the picture with the mechanisms in the present work?

I am a bit confused about the explanation in lines 211-213. The authors say that scaling up under-estimated costs is to deter overbidding. My understanding is that incentive compatibility has nothing to do with the predictions/the facilities do not see the predictions. Could the authors clarify?

What is the computational complexity of Auction 1? Vanilla facility location can at least be solved efficiently in practice via integer programming---is there a concise integer programming formulation for Auction 1?

**Ethical Concerns:**

["NO or VERY MINOR ethics concerns only"]

**Final Justification:**

I am in favor of accepting the paper since its contributions are substantial and technically interesting. (Also, I think that the ethics issue that was raised is a nonissue and should not work against this paper.)

**Limitations:**

Yes.

**Quality:**

3

**Strengths And Weaknesses:**

Strengths
-------------------

I think the problem and model studied are nice. Frankly, I think the model studied here is a more meaningful reflection of real-world procurement than anything modeled by prior work on strategic facility location :)

The improvement to the old result of Talwar on VCG's frugality ratio is a nice standalone contribution as well.

The paper is well written.

Weaknesses
-------------------

The robustness guarantees seem a bit weak since regardless of epsilon the frugality ratio is >= 5 which is worse than the VCG benchmark. Some result showing tightness would be nice here.

I know that frugality is a standard notion and has been studied before, but its motivation is a bit unclear to me. A sentence or two giving the intuition behind why the second best solution is the right measure would be helpful. To me it seems like a needless weakening of the strongest benchmark which is OPT.

It would be nice to include a few lines of intuitive description of how Theorem 5.1 is proved in the main body.

---

> ### Author Rebuttal · Authors · 2025-07-31
>
> Thank you for carefully reviewing our paper and providing constructive feedback. We address your questions below:
>
> * **Regarding the use of the frugal solution as the benchmark**
>
> Thank you for raising this point. The OPT benchmark is indeed stronger, but it is actually easy to verify that it is impossible to approximate. In fact, this impossibility is exactly what motivated the literature on frugality (starting with [1,2,3]), which introduced the frugal solution as a more appropriate benchmark. Therefore, this benchmark is not specific to facility location but, rather, the benchmark used in all of this literature on procurement auctions. Regarding the two papers that the reviewer referred to, note that Xu and Lu [IJCAI ‘22] do, in fact, also use this benchmark for their procurement auctions, while the Balcan et al. [NeurIPS ‘23] paper does not study procurement auctions.
>
> Maybe the simplest example to verify this inapproximability of OPT is to consider the case of just two sellers that provide the same service (e.g., in our case these could be two sellers that both have the same distance from every customer), so we need to procure the services of just one of them. If one of these sellers has a much higher cost than the other, then OPT would buy from the cheaper of the two at the cheaper cost, but this is impossible to approximate the payment in a truthful way in a setting without priors. This simple example captures the much more general fact that whenever the optimal solution is much better than the competition (resembling a “monopoly”), it is impossible to approximate it. We do briefly discuss this in the paragraph starting in line 43, but we agree that we would be happy to emphasize this further in the next revision. Also, we would be open to providing a short formal proof of the claim above if the reviewer believes this would make the need for the alternative benchmark more obvious.
>
> [1] Aaron Archer, Éva Tardos. Frugal path mechanisms, 2002
>
> [2] Kunal Talwar. The Price of Truth: Frugality in Truthful Mechanisms, 2003
>
> [3] Anna R. Karlin, David Kempe, Tami Tamir. Beyond VCG: Frugality of Truthful Mechanisms, 2005
>
>
> * **Regarding the coin flip between posting the predictions and using VCG**
>
> It is indeed true that if we knew we had perfect predictions regarding the opening costs, we could even achieve solutions with optimal cost (e.g., by identifying the optimal sellers based on these predictions and posting a take-it-or-leave-it price equal to their predicted cost). However, what is very important to emphasize here is that the predictions can actually be arbitrarily inaccurate (which is a key feature of the learning-augmented model), so this would lead to *unbounded robustness, even if we did that with very small probability $p$*.
>
> For example, consider the same simple instance that we discussed above: there are just two facilities, $A$ and $B$, where $A$ is OPT and $B$ is arbitrarily more costly than $A$. If the prediction regarding the cost of $A$ is even slightly higher than the true cost, then $A$ would reject the take-it-or-leave-it price, and the only feasible solution would be to buy the arbitrarily more expensive facility $B$. Note that this can readily be generalized to hold even if the mechanism is more conservative and posts a price (slightly) higher than the prediction, aiming to approximate OPT rather than achieve it exactly. Thus, even if we use VCG with the remaining probability of $1-p$, the expected payment can still be unbounded across the two cases, leading to unbounded robustness.
>
> In general, it is not difficult to achieve good consistency alone (even relative to the OPT benchmark) or good robustness alone (relative to the frugal benchmark). The true challenge (which is the focus of our submission) is to simultaneously achieve good robustness and consistency, without any knowledge regarding the prediction quality.
>
> * **I am a bit confused about the explanation in lines 211-213. The authors say that scaling up under-estimated costs is to deter overbidding**
>
> We agree that the use of the word "overbidding" in that context can be confusing and we would be happy to update  the wording accordingly.  Note that the threshold prices that are implied by Myerson's lemma, and which are used by all truthful mechanisms in single-parameter settings (like ours) effectively pay each bidder an amount equal to the "best lie" that they could report by overbidding and remaining a winner. One way to interpret this is that the mechanism pays the bidders the best amount they could earn by strategically overbidding so that they will not lie. As a result, if a mechanism chooses an allocation where some bidder has a lot of room to overbid and remain a winner, this directly implies that the amount that this bidder will have to be paid is high.
>
>
> * **Computation complexity of Auction 1 and integer program formulation**
>
> Although the focus of our submission is on the information limitations of truthful (learning-augmented) auctions, Auction 1 can indeed be implemented using integer programming (details provided below). We would be happy to include this in the next revision if the reviewer thinks it would add value.
>
> We can first solve a classic Uncapacitated Facility Location (UFL) integer program using the *predicted* opening costs to determine the predicted optimal set, $\hat{OPT}$. Once $\hat{OPT}$ is identified, its total modified cost can be calculated directly by scaling up the true cost of any of its facilities that were under-predicted. Subsequently, a second UFL integer program is solved using the *true* opening costs, but with an added linear constraint that explicitly rules out $\hat{OPT}$ as a feasible solution,
>
> Specifically, if $y_l$ is the binary variable that equals 1 if facility $l$ is opened and 0 otherwise, the constraint that rules out $\hat{OPT}$ can be written as:
>
> $\sum_{l \in \hat{OPT}} (1 - y_l) + \sum_{l \notin \hat{OPT}} y_l \ge 1$
>
> This allows the integer program to find the optimal solution amongst all feasible sets *excluding* $\hat{OPT}$, thereby finding the best alternative outcome. The final output of the auction is then determined by comparing the modified cost of $\hat{OPT}$ with the cost of the best alternative solution; the set with the lower cost is chosen.
>
> Finally, the threshold payment of each facility in the output set can be computed by running a UFL integer program without the facility.

---

> > ### Comment · Reviewer_ENCf · 2025-08-03
> >
> > Thank you for your detailed response.

---

### Official Review · Reviewer_xSdS · 2025-06-30

**Clarity:** 2
**Significance:** 3
**Originality:** 3
**Rating:** 2
**Confidence:** 5

**Summary:**

The paper studies procurement auctions for the strategic uncapacitated facility location (UFL) problem, where facilities are owned by strategic agents with private opening costs. The goal is to open facilities and determine payments to minimize total cost, balancing opening and user connection costs. The authors evaluate auction performance using the frugality ratio, which compares auction cost to a “second-best” benchmark. They first show that the classical VCG auction has a tight frugality ratio of 3. Then, they introduce learning-augmented auctions that use potentially inaccurate cost predictions to significantly reduce payments when predictions are accurate, while still maintaining bounded frugality under adversarial errors. They also propose an error-tolerant auction variant that adapts to approximately accurate predictions, achieving improved frugality guarantees as a function of prediction error.

**Questions:**

Just wondering if the authors have thought about this comment:

**Furthermore, the prediction-augmented algorithm may have limited practical applicability, as it penalizes certain agents by scaling up their costs based on potentially inaccurate predictions. The design appears somewhat single-minded, focusing narrowly on optimizing a specific objective (i.e., frugality) while overlooking broader consequences of the mechanism, such as fairness or unintended distortions in participation incentives.**

And if they have done some simulations to see how biases in prediction can hurt some of the players?

**Ethical Concerns:**

["Major Concern: Discrimination, bias, and fairness"]

**Final Justification:**

I am not fully convinced by the authors' response regarding the following

> Furthermore, the prediction-augmented algorithm may have limited practical applicability, as it penalizes certain agents by scaling up their costs based on potentially inaccurate predictions. The design appears somewhat single-minded, focusing narrowly on optimizing a specific objective (i.e., frugality) while overlooking broader consequences of the mechanism, such as fairness or unintended distortions in participation incentives.

Furthermore, the exposition needs some additional work; the paper is not well written.

**Limitations:**

See my earlier comments.

**Paper Formatting Concerns:**

See my comments in "Questions" regarding fairness, the impact of biases in the prediction and that the algorithm is single-minded.

**Quality:**

1

**Strengths And Weaknesses:**

**Strengths:**
The paper addresses an important and timely problem---the design of procurement auctions for strategic uncapacitated facility location---by exploring how predictive information can improve frugality in mechanism design.

**Weaknesses:**
The paper is poorly written and lacks clarity in both exposition and notation. Key concepts, such as the VCG auction, are not properly defined when first introduced. The proofs are difficult to follow, partly due to inconsistent and undefined notation. For example, the term $U_{\ell, f}$ appears in key arguments without any formal definition, and the notation for the auction mechanism $\mathcal{M}$ is used inconsistently throughout the paper with varying input sets, without acknowledgment. Similarly, $\mathsf{OPT}$, a central object in the definition of the frugality ratio, is used before being clearly defined. Additionally, the motivation for focusing on the VCG auction and for using the frugality ratio as the primary performance metric is not clearly justified.

Furthermore, the prediction-augmented algorithm may have limited practical applicability, as it penalizes certain agents by scaling up their costs based on potentially inaccurate predictions. The design appears somewhat single-minded, focusing narrowly on optimizing a specific objective (i.e., frugality) while overlooking broader consequences of the mechanism, such as fairness or unintended distortions in participation incentives.

Finally, the paper completely overlooks the computational aspects of the optimization problems involved in the proposed mechanisms. These problems are often NP-hard, and the paper provides no discussion on tractability or practical solvability.


These are just some example problems. The paper has some potential. But, in its current form, it is not ready for publication.

---

> ### Author Rebuttal · Authors · 2025-07-31
>
> Thank you for carefully reviewing our paper and providing constructive feedback. We apologize for omitting the definition of  $U_{\ell, f}$, which denotes the set of users assigned to $\ell$ in OPT and to $f$ in the frugal solution $F$. We understand that this omission (which, unsurprisingly, all the reviewers pointed out) may have made it harder to parse the proofs. We also agree that there is some room for improvement in the writing, but we believe that these issues are easy to address (e.g., by just adding the definition of $U_{\ell, f}$) and, as we point out in the rest of the rebuttal, some of the reviewer’s concerns are actually already addressed in the initial submission. We therefore believe that the overall assessment of our submission is somewhat harsh, and we would be happy to use the discussion period as an opportunity to convince the reviewer that this is the case, and that we can address all of their concerns in the next revision of the paper.
>
>
> * **OPT, a central object in the definition of the frugality ratio, is used before being clearly defined**
>
> This is actually not accurate. The first time that OPT is used is on line 129 where it is defined both in words as the optimal facility set as well as with a formal mathematical definition as $\arg\min_{S \subseteq L} c(S)$.
>
>
> * **Key concepts, such as the VCG auction, are not properly defined when first introduced**
>
> The formal definition of the VCG auction is at the bottom of page 4, at the very beginning of Section 3, right after the preliminaries. If the reviewer prefers that we add a pointer to its definition when the VCG mechanism is first mentioned in the introduction, we would be happy to do so. If there are other key concepts that are not properly defined when first introduced, we would appreciate it if the reviewer could provide us with additional examples
>
>
> * **The notation for the auction mechanism is used inconsistently throughout the paper with varying input sets**
>
> The mechanism notation is introduced at the bottom of page 3 and in its full generality it takes as input the tuple $(U,b,d)$. One line after its first definition, we also state that we often just provide $(b)$ as its input when $U$ and $d$ are clear from context. Then, at the top of page 4, we define truthfulness and point out that whenever a mechanism is truthful, it is a dominant strategy for bidders to submit the truth, so that $b=o$. As a result, since the whole paper is restricted to truthful mechanisms, we can also directly use $o$ instead of $b$ as the input of the mechanism.
>
> Does this address the reviewers’ concerns, or is there some other inconsistency that they have in mind?
>
>
> * **The design appears somewhat single-minded, focusing narrowly on optimizing a specific objective (i.e., frugality) while overlooking broader consequences of the mechanism, such as fairness or unintended distortions in participation incentives.**
>
> When designing procurement auctions, the most obvious and natural objective to optimize is the cost of the buyer who will be using this auction. This is analogous to the design of (forward) auctions, where the seller running the auction aims to maximize revenue. As a result, it is a central goal in the mechanism design literature to minimize cost and maximize revenue when designing such auctions. We agree with the reviewer that an important secondary objective would be to understand the extent to which the buyer’s cost can be optimized subject to fairness constraints. In fact, although we have not conducted any simulations regarding such fairness considerations, our mechanisms always guarantee individual rationality (i.e., they ensure that no bidder will ever receive a payment less than the cost that they report), which is a classic form of a participation incentive in mechanism design. Note, however, that in the setting that we consider even the primary objective of cost minimization was previously not well understood, so this is the most natural first goal.
>
> On a related note, we noticed that the reviewer flagged our submission for major ethical concerns, based on potential discrimination, bias, and fairness issues. To draw an analogy, the same concerns that the reviewer is raising here would directly apply to the classic (Nobel award winning) work of Myerson on the design of revenue-optimal auctions. Specifically, this work shows that the optimal auction for maximizing revenue when the value of each bidder is drawn from a distribution is to post bidder-specific posted prices, which is precisely the type of concern that the reviewer is raising. Although we would be happy to add a discussion regarding fairness considerations, including the participation incentives discussed below, we believe it is unreasonable for the reviewer to use this as grounds for rejection.
>
>
> * **Regarding computational considerations**
>
> Although the focus of our submission is on the information limitations of truthful (learning-augmented) auctions can be implemented using integer programming (see our response to Reviewer ENCf for more details)

---

> > ### Comment · Reviewer_xSdS · 2025-08-01
> >
> > I thank the authors for their response.
> >
> > I still think the way $\mathcal M$ is used in the paper is confusing as it has a varying number of inputs. See Lines 131, 143, and 146 as examples. This can be fixed though in the revision.
> >
> > More importantly, can you explain how you can address/mitigate this issue?
> >
> > "*The design appears somewhat single-minded, focusing narrowly on optimizing a specific objective (i.e., frugality) while overlooking broader consequences of the mechanism, such as fairness or unintended distortions in participation incentives.*"
> >
> > And finally, the answer to my "computational concerns" is not very convincing.

---

> > > ### Author Response · Authors · 2025-08-04
> > >
> > > * **Regarding the mechanism $M$ notation**
> > >
> > > We would be happy to address any issues with the way the notation of mechanism $M$ is used, but we are unfortunately still unsure about what issues the reviewer is referring to. Specifically, the lines that the reviewer is referring to are all consistent with the explanation that we provided in the rebuttal: a (non-learning-augmented) mechanism takes as input $(U,b,d)$ and a learning-augmented mechanism also takes a prediction $\hat{o}$ as one additional input. For truthful mechanisms, $b$ can be replaced with $o$, and the dependence on $U$ and $d$ is dropped whenever clear from context, for readability purposes.. All of this notation is standard in the (learning-augmented) mechanism design literature, so it would be helpful if the reviewer could provide us with additional guidance regarding the changes that they would like us to make in the revision. Would it be sufficient for us to emphasize the fact that we can use $b$ and $o$ interchangeably when it comes to truthful mechanisms?
> > >
> > >
> > > * **Regarding the reviewer’s claim that our design appears to be “single-minded, focusing narrowly on optimizing a specific objective”**
> > >
> > > As we tried to convey in our rebuttal, we actually believe this criticism is not well-justified: the objective we are optimizing (i.e., the final cost paid by the buyer) is by far the most important one for the buyer (who is the one designing/choosing the auction), not only in theory, but also in practice, and it was not well understood in prior work. Therefore, unless the reviewer believes that our results are technically weak or insufficient, we do not see why our focus is narrow.
> > >
> > > Having said that, there are some specialized settings (e.g., procurement auctions led by government agencies), where fairness and social welfare considerations can also play a role, or impose constraints on how the auction is designed. Even though these settings are the exception rather than the norm, and even though minimizing the cost is the central objective there too, these applications provide some motivation for follow-up work to further refine our results along this direction. We would, therefore, be happy to add a paragraph somewhere in our submission (e.g., we could introduce a conclusion section) that discusses these considerations, the impact that our proposed auction can have along these lines, as well as the role that the parameter $\epsilon$, chosen by the designer, can play (e.g., this directly affects the extent of the scaling). We could also share some thoughts regarding the obstacles that may arise when trying to optimize robustness and consistency subject to constraints that limit or disallow such scaling. An obvious obstacle that arises is that in order to achieve non-trivial consistency guarantees a mechanism needs to use the predictions, and this inevitably leads to what the reviewer refers to as bias or distortion. We could also discuss how the use of prior information is standard both in auction theory and in practical auction design. For instance, as we pointed out in our rebuttal, the classic work of Myerson on revenue-optimal auctions uses bidder-specific reserve prices based on prior distributional information, and the way that many real-world auctions, such as the ad-auctions that companies like Google and Meta run daily at a massive scale, leverage vast amounts of historical information to determine both the allocation and the pricing. Would a paragraph along these lines sound reasonable?
> > >
> > > * **Regarding computational considerations**
> > >
> > > Finally, even though the reviewer did not provide a justification regarding why they found our response on computational considerations unconvincing, we would like to add the following: although we do think that the theoretical analysis of algorithms from a computational standpoint is interesting, we believe that the information limitations that arise in mechanism design settings, like the one that we study, are much more relevant and binding in practice. For example, for most real world applications, the size of the instances would be small enough that even a naive brute force algorithm (e.g., one that exhaustively considers all possible subsets of facility locations) could be tractable. Solving an integer program like the one we proposed can, in fact, solve significantly larger instances in quite a reasonable amount of time. What makes computational considerations even less binding in this setting is that, unlike ad-auctions, that are run repeatedly and need to be very fast, the procurement applications that motivate our work are run very infrequently, so time is not the scarce resource here. This is in stark contrast to the information limitations that arise even for very small instances, requiring a careful design to align the bidders’ incentives with those of the designer.

---

### Official Review · Reviewer_Efkg · 2025-07-03

**Clarity:** 2
**Significance:** 3
**Originality:** 3
**Rating:** 5
**Confidence:** 2

**Summary:**

This paper studies the strategic uncapacitated facility location problem (UFL). In UFL, an agency wants to procure a set of facilities among a set of potential locations to provide service to its users. Each facility location comes with an individual cost. The goal is to choose a set of facility locations that minimizes the sum of the distances from users to facilities and the total payment to the facility owners.

In strategic UFL, each facility owner has a private cost, and the agency does not know these private values. Instead, it runs an incentive-compatible auction to determine which locations to procure and how much to pay. In the literature, there has been some work on analyzing the worst-case guarantees of IC auctions, comparing them to the total cost when the location costs are public. More formally, the benchmark is the frugality ratio, which is the worst-case ratio between the total cost of a truthful auction and the cost of the second-best solution when the agency knows the true costs.

The first result of the paper is a tight analysis of the frugality ratio for the VCG auction. Previously, the best known upper bound for VCG was 4. This work improves it to 3, with a matching lower bound instance.

The second contribution is that while most previous works focus only on worst-case guarantees, this paper designs and analyzes new auction frameworks that take into account predictions on the cost of each location. These mechanisms not only perform well when the predictions are accurate, but also retain worst-case guarantees when the predictions are wrong.

**Questions:**

The notation $U_{\ell, f}$ is not defined. Is it the set of users that are assigned to $\ell$ in OPT and assigned to $f$ when the facility set is F?

**Ethical Concerns:**

["NO or VERY MINOR ethics concerns only"]

**Final Justification:**

I recommend acceptance of this submission. Though the presentation could use some additional work, I believe the paper addresses an interesting and important problem and would be valuable to this venue.

**Limitations:**

yes

**Quality:**

3

**Strengths And Weaknesses:**

Strengths:
1.The problem is well-motivated, and the authors provide a tight analysis of the VCG auction, improving upon previous results in the literature.
2. The paper goes beyond traditional worst-case analysis. It designs algorithms that achieve strong guarantees when given accurate predictions, while still providing baseline performance guarantees even when the predictions are completely inaccurate.

Weaknesses:
1. The presentation can be improved—some proofs are hard to follow, with a missing definition that makes them difficult to parse.
2. The model is somewhat limited: it focuses on the case where the agency has accurate predictions for the cost of each location. It would be more interesting to consider Bayesian models, where the true costs are drawn from known prior distributions.

---

> ### Author Rebuttal · Authors · 2025-07-31
>
> Thank you for carefully evaluating our paper and for providing us with constructive feedback. We apologize for omitting the definition of $U_{\ell, f}$, which is exactly the one you mentioned. We have updated the submission to include this definition. We briefly address you comment regarding the Bayesian model below.
>
> * **Regarding the learning-augmented model and its comparison to Bayesian models**
>
> Both the Bayesian model and the learning-augmented model are part of the same broader literature, aiming to move beyond worst-case analysis, however there are important differences between the two. The most important among them is that in the learning-augmented model the quality of the predictions is unknown and they can actually be arbitrarily inaccurate (this is in contrast to the reviewer's claim that the learning-augmented model "focuses on the case where the agency has accurate predictions"). On the other hand, the Bayesian model actually assumes that the distributions that the agency is provided with are (perfectly) accurate, and the algorithms and mechanisms are then evaluated in expectation over that randomness. In settings where such accurate distributional information is easy to extract, we agree that the Bayesian models are quite interesting and relevant. However, in settings where such information may be harder to gather and the predictions of the machine learning algorithms are less dependable, we believe that the learning-augmented model is more relevant. Another signal regarding the significance of the learning-augmented model is the fact that more than 250 papers have focused on this model during the last 5 years according to the "Algorithms with Predictions" github.

---

> > ### Comment · Reviewer_Efkg · 2025-08-04
> >
> > Thank you for the response!

---

### Note · Authors · 2025-08-13

We would like to thank the reviewers for the time they took to review our submission, to participate in the subsequent discussion, and to ask helpful follow-up questions. We believe that their feedback has already helped us to significantly improve the submission and we hope that they agree that our responses addressed all of their concerns.

---

### Decision · Program_Chairs · 2025-09-17

**Decision:**

Accept (poster)

**Comment:**

This paper studies the strategic uncapacitated facility location problem (UFL). The first result of the paper is a tight analysis of the frugality ratio for the VCG auction. Previously, the best known upper bound for VCG was 4. This work improves it to 3, with a matching lower bound instance. The second contribution is that while most previous works focus only on worst-case guarantees, this paper designs and analyzes new auction frameworks that take into account predictions on the cost of each location. These mechanisms not only perform well when the predictions are accurate, but also retain worst-case guarantees when the predictions are wrong.

The rebuttal appears to be able to address the main issues about presentation. I also suggest the authors to add some word about the complexity of the ILP algorithm. Anyway these issue are very minor. On the other side the paper concerns a well-studied problem, and presents tight and relevant results. Hence, we here support acceptance of this paper.